**DOI: 10.1038/ncomms14378**　**OPEN**

# A microRNA-initiated DNAzyme motor operating in living cells

Hanyong Peng[1], Xing-Fang Li[1], Hongquan Zhang[1] & X. Chris Le[1]

Synthetic DNA motors have great potential to mimic natural protein motors in cells but the operation of synthetic DNA motors in living cells remains challenging and has not been demonstrated. Here we report a DNAzyme motor that operates in living cells in response to a specific intracellular target. The whole motor system is constructed on a 20 nm gold nanoparticle (AuNP) decorated with hundreds of substrate strands serving as DNA tracks and dozens of DNAzyme molecules each silenced by a locking strand. Intracellular interaction of a target molecule with the motor system initiates the autonomous walking of the motor on the AuNP. An example DNAzyme motor responsive to a specific microRNA enables amplified detection of the specific microRNA in individual cancer cells. Activated by specific intracellular targets, these self-powered DNAzyme motors will have diverse applications in the control and modulation of biological functions.

[1] Division of Analytical and Environmental Toxicology, Department of Laboratory Medicine and Pathology, Faculty of Medicine and Dentistry, University of Alberta, 10-102 Clinical Sciences Building, Edmonton, Alberta, Canada T6G 2G3. Correspondence and requests for materials should be addressed to H.Z. (email: hongquan@ualberta.ca) or to X.C.L. (email: xc.le@ualberta.ca).

Cells use protein motors to transport molecules and organelles along cytoskeleton tracks, allowing a high degree of spatial and temporal organization of cellular molecules and organelles[1–3]. Protein motors require energy to accomplish intracellular transport along specific tracks[4]. For instance, three well-known protein motors, myosin, kinesin and dynein, use the energy from hydrolysis of ATP to perform their autonomous and processive movement along actin filaments or microtubules[5,6].

Inspired by protein motors, researchers have recently constructed various synthetic DNA motors to mimic the functions of protein motors[7–12]. The remarkable specificity and predictability of Watson–Crick base pairing make DNA an appealing construction material to build the synthetic motor systems[13–16]. Construction of DNA motors has focused on three main aspects: exploiting various fuel energies, improving the design of DNA tracks and controlling the direction of walking. DNA hybridization[17,18], hydrolysis[19] and conformational transition[20] have been used to power the DNA motors[21,22]. One-dimensional tracks were initially used to demonstrate the concept of DNA motors[23–25]. DNA origami was further used to construct two-dimensional tracks, increasing the walking steps and directionality of the motor[26–28]. Recently, DNA tracks were built on nano- and micro-materials, including gold film[7], microparticles[8], carbon nanotubes[10,29] and gold nanoparticles (AuNPs)[30,31], further improving the mobility and processivity of the DNA motors. Instructional strands were included precisely at specific positions of DNA tracks, enabling control of the walking direction of the motors[22,32].

Although various synthetic DNA motors have been tested *in vitro*, the ultimate goal of introducing them into cells to perform specific biological functions has not yet been achieved[33–35]. To operate in living cells, synthetic DNA motors must confront the following challenges: first, all components of the motor system, including the motor and its track, must be readily taken up by living cells; second, the operation of the motor must be initiated by specific molecules within the cells; third, the motor has to be self-powered to enable autonomous intracellular walking, because external addition of fuel DNA strands or protein enzymes is not desirable; finally, the operation of the motor in living cells should be monitored in real time.

We describe a strategy of constructing a DNAzyme motor system on a AuNP that meets all four requirements. Integration of all motor components onto a single AuNP facilitates cellular uptake of the DNA motor system. The operation of the DNAzyme motor can be initiated by a specific molecule in the cell, for example, microRNA (miRNA). The operation of the DNAzyme motor is self-powered, because the autonomous walking of the motor along the AuNP is fueled by DNAzyme-catalysed substrate cleavage. Each step of the DNAzyme movement results in the cleavage of a substrate strand and the release of a fluorescently labelled DNA strand from the AuNP. Thus, each walking step of the motor results in an increase in fluorescence, allowing real-time imaging of the intracellular operation of the motor.

## Results

### Principle of intracellular target-initiated DNAzyme motor.

Figure 1 depicts the overall concept and the intracellular operation of a DNAzyme motor. The motor system is constructed on a functionalized AuNP onto which are conjugated hundreds of substrate strands and dozens of DNAzyme molecules that are each silenced by a locking strand (Supplementary Fig. 1). The locking strand is designed to respond to a specific intracellular target. As a proof of principle, we chose a specific miRNA as the cellular target. For imaging purposes, we fluorescently labelled the locking strand with cyanine5 (Cy5) and the substrate strand with carboxyfluorescein (FAM). When the DNAzyme motor is inactive, the fluorescence from both Cy5 and FAM is quenched by the AuNP.

Once the DNAzyme motor is taken up by the cells, the intracellular miRNA hybridizes with the locking strand through a strand-displacement reaction, releasing the locking strand from the DNAzyme. The unlocked DNAzyme then hybridizes to its substrate on the AuNP. In the presence of the cofactor $Mn^{2+}$, the DNAzyme cleaves a substrate molecule, releasing the FAM-labelled segment. Cleavage of the DNA–RNA chimeric substrate provides the energy needed for the DNAzyme to move from one substrate strand to the next, achieving the autonomous and processive walking along the AuNP. Each walking step and substrate cleavage is accompanied by the release of the fluorescently labelled segment of the substrate. As these molecules are detached from the AuNP, they become fluorescent. Monitoring these fluorescent molecules enables real-time detection of the intracellular motion of the DNAzyme motor.

The substrate strand (sequence in Supplementary Table 1) is a DNA–RNA chimeric sequence composed of an RNA nucleotide flanked by two DNA domains. These two DNA domains are binding regions of two arms of the DNAzyme motor. To enhance the accessibility of the substrate strand to the DNAzyme, we added a 14-thymine (T) spacer **S1** to the substrate at the 5′-end that is conjugated to the AuNP. The 3′-end of the substrate is labelled with a FAM molecule whose fluorescence is quenched by the AuNP.

The DNAzyme, a truncated form of 8–17E DNAzyme[36], consists of a catalytic core sequence flanked with binding **Arm 1** and **Arm 2** (Supplementary Fig. 1). The DNAzyme is conjugated to the AuNP through a single-stranded spacer **S2** linked to the 3′-end of **Arm 2**. The spacer **S2** comprises a 42-thymine domain that is conjugated to the AuNP and provides the spatial distance needed for the motor walking. A 16 nt domain **T*1** and **Arm 2** form the locking region. The sequence selection of domain **T*1** depends on the specific molecules designed to initiate the motor operation. For example, to construct a DNAzyme motor that is initiated by specific intracellular miRNA, we designed a locking strand that contains a target-binding domain complementary to the target miRNA (miR-10b) and a sequestering domain complementary to **Arm 2**. The hybridization of the locking strand to the domain **T*1** and **Arm 2** forms a duplex with a 7 nt toehold at the 3′-end of the locking strand, which sequesters **Arm 2** from binding to the substrate strands on the track. With the locking strand hybridized to the DNAzyme, the DNAzyme motor is inactive. It is the inactive DNAzyme motor that is introduced to living cells and subsequently switched on by the specific cellular target. When the inactive DNAzyme motor interacts with the target molecule, for example, miRNA, the target miRNA can hybridize with the locking strand through a strand displacement reaction, exposing **Arm 2** and initiating the operation of the DNAzyme motor (Supplementary Fig. 2).

Labelling a Cy5 molecule at the 3′-end of the locking strand enables the detection of the intracellular location of the target molecule. When the locking strand hybridizes to the DNAzyme strand, the fluorescence of the Cy5 molecule is quenched by the AuNP. However, when the target miRNA forms a duplex with the locking strand through the strand displacement reaction, the locking strand is removed from the AuNP surface. Thus, the fluorescence of the Cy5 molecule in the duplex is restored, signalling the location of the target miRNA.

To ensure efficient sequestering of the DNAzyme and high mobility of the motor after initiation, we designed **Arm 1** and

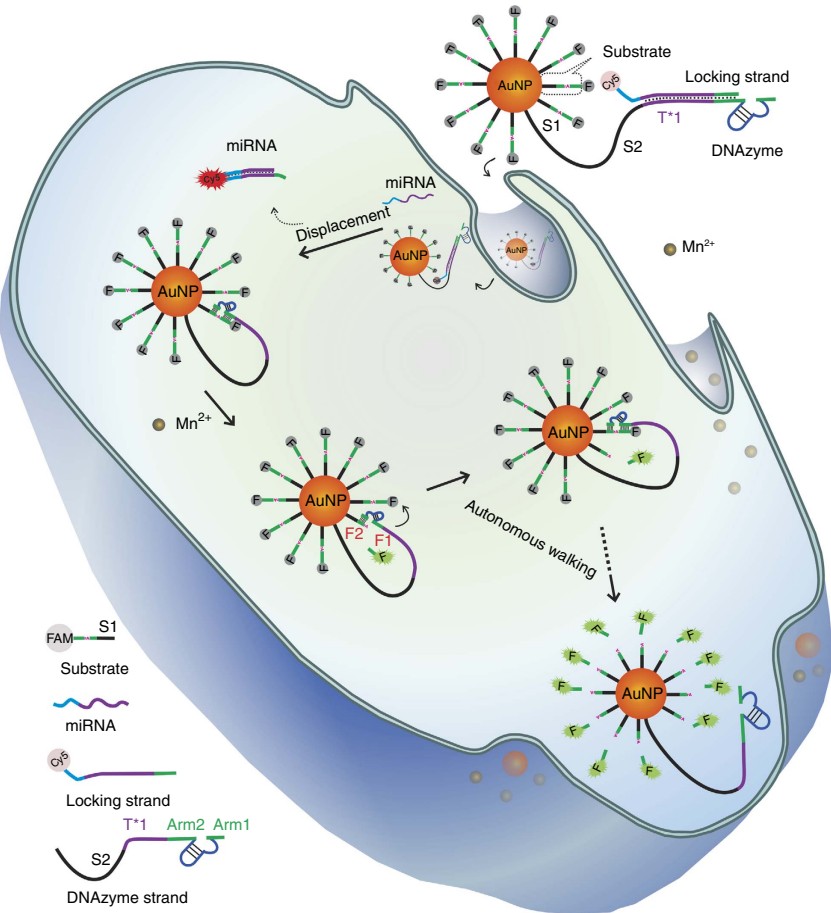

**Figure 1 | Intracellular operation of a DNAzyme motor initiated by a specific miRNA.** The DNAzyme motor system is constructed on the AuNP. The AuNP is functionalized with hundreds of substrate strands and dozens of DNAzyme molecules that are each silenced by a locking strand. Inside the cells, the target miRNA hybridizes to the locking strand and releases the locking strand from the DNAzyme through a strand displacement reaction. The unlocked DNAzyme subsequently hybridizes to its substrate. The cofactor $Mn^{2+}$ activates the DNAzyme, which cleaves the substrate, generating two DNA segments **F1** and **F2**. The FAM-containing **F1** segment is released from the AuNP surface, restoring its fluorescence that is previously quenched by the AuNP. Meanwhile, the DNAzyme dissociates from **F2** and subsequently hybridizes to the next substrate strand, achieving the walking of the motor from one substrate strand to the next. This stepwise walking is repeated autonomously, driving the DNAzyme motor to traverse along the AuNP surface. Monitoring the fluorescence of FAM provides real-time imaging of the intracellular operation of the motor in live cells.

**Arm 2** to contain 5 and 7 nt, respectively. A 7 nt **Arm 2** allows the locking strand to occupy most or all nucleotides of **Arm 2**, thus efficiently inhibiting **Arm 2** from binding to substrate strands on the AuNP. In addition, when the duplex between the target miRNA and the locking strand is formed through the strand displacement reaction, the duplex can be efficiently dissociated from **Arm 2**, because the hybrid ($<7$ nt) between **Arm 2** and the sequestering domain is unstable. Importantly, the 7 nt **Arm 2** is sufficient for the DNAzyme to form a stable complex between the DNAzyme and the substrate strand conjugated on the same AuNP. Confining the DNAzyme and its substrate on a 20 nm AuNP surface leads to a high local concentration of DNAzyme and substrate, enhancing their hybridization. The estimated melting temperature ($T_m$) of the hybrid of the DNAzyme with its substrate on the same AuNP is 41 °C. Therefore, when the locking strand is liberated by the target miRNA, the DNAzyme motor is available to interact with a substrate stand on the AuNP track. In the presence of divalent metal cofactors, such as $Mn^{2+}$, the DNAzyme cleaves the substrate at the single-ribonucleotide junction, generating two DNA segments **F1** and **F2**. The FAM-containing **F1** then dissociates from **Arm 2** and escapes from the AuNP surface,

and therefore its fluorescence is restored. As **Arm 1** contains only 5 nt, the remaining hybrid between **Arm 1** and **F2** (the substrate segment attached to AuNP) becomes unstable ($T_m = 7$ °C). The DNAzyme therefore dissociates from **F2** and subsequently hybridizes to the next substrate strand, achieving the walking of the DNAzyme motor from one substrate strand to the next. This stepwise walking, fueled by DNAzyme-catalysed cleavage of the substrate, is repeated, driving the DNAzyme motor to traverse along the AuNP surface (Supplementary Fig. 2). Similar to other DNA motors that use DNA tracks built on nano- and micro-materials[8,30,31], walking of the DNAzyme motor along the AuNP is stochastic.

Each walking step of the DNAzyme motor releases one **F1** from the AuNP surface, restoring the fluorescence of the FAM molecule in **F1**. Therefore, the increase in fluorescence corresponds to the number of steps that DNAzyme motors have moved during the specific operating time. The intracellular operation of the DNAzyme motor can be imaged in real time by measuring the fluorescence increase. In addition, the fluorescence increase is proportional to the amount of the target miRNA strand in the cell, enabling *in situ* amplified detection of miRNA in living cells.

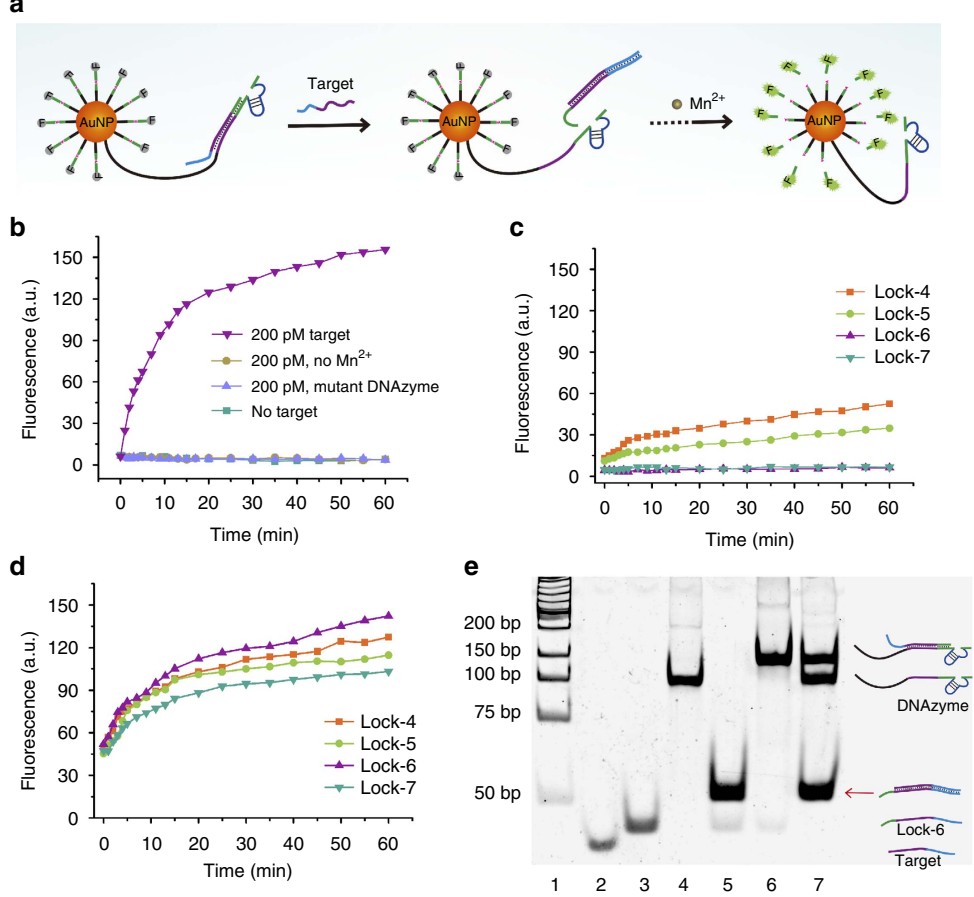

**Figure 2 | Evaluation of a DNAzyme motor for its performance in solution.** (**a**) Operation of DNAzyme motor initiated by the specific nucleic acid target. (**b**) Real-time monitoring of fluorescence generated by the DNAzyme motor in response to the target sequence and cofactor $Mn^{2+}$. The relative s.d. (RSDs) from replicate experiments were 2.1–5.4%. (**c**) Background fluorescence generated by four motors that vary by the locking strand. RSDs were 3.5–7.2%. (**d**) Real-time monitoring of fluorescence generated by the four DNAzyme motors in response to 200 pM target sequence. RSDs were 4.1–8.3%. (**e**) Gel images showing strand displacement of the DNAzyme strand by the target DNA, forming a duplex between the target and the locking strand. Lane 1: DNA ladder; lane 2: target DNA; lane 3: locking strand; lane 4: DNAzyme strand; lane 5: mixture of locking strand and target DNA at 1:1 molar ratio; lane 6: mixture of locking strand and DNAzyme strand at 1:1 molar ratio; lane 7: mixture of locking strand–DNAzyme strand complex and target DNA at 1.1:1 molar ratio. The strong band marked a red arrow corresponds to the hybrid between the locking strand and the target strand, resulting from the strand displacement reaction.

**Evaluation of the DNAzyme motor in test tubes**. To facilitate the optimization and evaluation, we used a DNA strand with the same sequence as miR-10b to initiate the operation of the DNAzyme motor (Fig. 2). In the absence of a target sequence, **Arm 2** of the DNAzyme is sequestered by the locking strand and the DNAzyme motor is inactive, which is demonstrated by no observable fluorescence increase over the operating time (Fig. 2b). On addition of 200 pM target sequence, the target-triggered strand displacement exposes **Arm 2** of the DNAzyme, thereby enabling the DNAzyme to hybridize to one substrate strand on the track. In the presence of the cofactor $Mn^{2+}$, the DNAzyme is activated to cleave the substrate strand, initiating autonomous walking of the motor along AuNP. The continuous fluorescence increase over the 60 min operating time suggests the autonomous walking of the motor. When the cofactor is absent, no fluorescence increase is observed, confirming that the operation of the DNAzyme motor relies on both the specific target sequence and the cofactor for the DNAzyme.

To examine the function of the DNAzyme, we mutated two nucleotides (G to T) of its catalytic core and used this mutant DNAzyme sequence to construct a mutant DNAzyme motor system. No fluorescence increase was observed (Fig. 2b) from the incubation mixture of the mutant motor system, 200 pM target sequence and 0.5 mM cofactor $Mn^{2+}$. These results further support that the fluorescence increase of the functional DNAzyme motor results from its selective response to the target sequence and rule out the possibility of target-independent substrate degradation (for example, RNase cleavage). The fluorescence results are also consistent with results from gel electrophoresis (Supplementary Fig. 3).

The locking strand is used to silence the DNAzyme and respond to target miRNA. An effective locking strand should completely sequester **Arm 2** of the DNAzyme and efficiently expose it in response to the specific target. Incomplete sequestering of **Arm 2** would result in unwanted target-independent operation of the DNAzyme motor, whereas inefficient exposing of **Arm 2** by target miRNA would decrease the sensitivity of the motor. We designed four locking strands (Lock-4, 5, 6 and 7; Supplementary Table 1) consisting of a conserved target-binding domain and a sequestering domain with different lengths: 4, 5, 6 and 7 nt, respectively. We used these four locking strands to construct the DNAzyme motor and compared

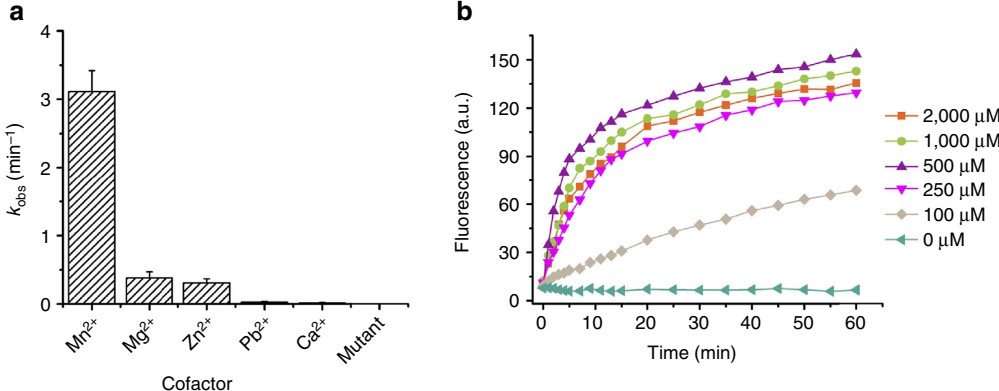

**Figure 3 | Effect of the cofactors on the operation of the DNAzyme motor.** (**a**) Multiple-turnover cleavage rate ($k_{obs}$) of the DNAzyme motor tested using 200 pM target DNA and different divalent metal ions as cofactors. The concentrations of metal ions were 0.5 mM $Mn^{2+}$, 10 mM $Mg^{2+}$ and $Ca^{2+}$, 0.01 mM $Zn^{2+}$ and 0.2 mM $Pb^{2+}$. A mutant DNAzyme motor (sequence in Supplementary Table 1) was also tested using 200 pM target DNA and 0.5 mM $Mn^{2+}$. Error bars represent 1 s.d. from duplicate experiments. (**b**) Real-time fluorescence generated by the DNAzyme motor that was initiated by 200 pM target DNA and activated by different concentrations of the cofactor $Mn^{2+}$. The RSDs were 2.4–6.6%.

their efficiencies of sequestering and target response. Good efficiency of sequestering, as indicated by negligible background, is achieved using sequestering domains of 6 and 7 nt (Fig. 2c and Supplementary Fig. 4). The highest efficiency in initiating the operation of the DNAzyme motor, as indicated by the fluorescence intensity, is seen when Lock-6 is used in the motor system (Fig. 2d). We then used gel electrophoresis to test the strand displacement efficiency of Lock-6 (Fig. 2e). In lane 7, the presence of a strong band corresponding to the hybrid of Lock-6 with the target sequence and the absence of a band corresponding to the single-stranded target sequence further support that the use of Lock-6 allows for complete displacement of the DNAzyme strand by the target sequence. To explain why Lock-6 is a better locking strand than Lock-7, we estimated the Gibbs free energy ($\Delta G$) of hybridization of Lock-6 and Lock-7 with the DNAzyme strand and target miRNA. The $\Delta G$ values of hybridization of Lock-6 and Lock-7 with the target miRNA are the same, $-30.6$ kcal mol$^{-1}$, whereas the $\Delta G$ values of hybridization of Lock-6 and Lock-7 with the DNAzyme strand are $-27.1$ and $-28.7$ kcal mol$^{-1}$, respectively. Thus, the $\Delta G$ of hybridization between Lock-7 and the target miRNA is slightly higher than that of hybridization between Lock-7 and the DNAzyme strand, consistent with the decreased strand displacement efficiency.

The metal cofactors are usually required to achieve the catalytic activity of the DNAzyme[37,38]. We tested the operation of the DNAzyme motor in response to 200 pM target sequence and different divalent metal ions. Although $Pb^{2+}$ is the most effective cofactor for the original 8–17E DNAzyme, we found that $Mn^{2+}$ was the best cofactor for the truncated form of the DNAzyme. (**Arm 1** and **Arm 2** of the original 8–17E DNAzyme were truncated from 9 to 5 and 7 nt, respectively.) The multiple turnover cleavage rate $k_{obs}$ follows the order of $Mn^{2+} > Mg^{2+} > Zn^{2+} > Pb^{2+} > Ca^{2+}$ (Fig. 3a), which is consistent with results from gel electrophoresis (Supplementary Fig. 5). We further compared single turnover cleavage rates of DNAzyme when using $Mn^{2+}$ and $Mg^{2+}$ as the cofactor. Similarly, $Mn^{2+}$ resulted in a stronger catalytic activity than $Mg^{2+}$ (Supplementary Fig. 6). We then examined the effect of $Mn^{2+}$ concentration on the operation of the DNAzyme motor. The DNAzyme motor operates reliably in the presence of 250 to 2,000 μM $Mn^{2+}$ (Fig. 3b, Supplementary Figs 7 and 8). Therefore, intracellular operation of the DNAzyme motor is feasible, because cellular uptake of hundreds of μM $Mn^{2+}$ does not

impact the viability of cells[39]. We also tested the operation of the DNAzyme motor under different pH conditions. The motor showed reliable performance in the pH range from 7.0 to 9.0 (Supplementary Fig. 9).

We examined the specificity of the DNAzyme motor by testing five variants of single-base mismatch. These five variants were designed to have the mismatch base at different representative sites. The fluorescence increase resulting from the 200 pM target sequence is significantly larger than those increases from the five variants at the same concentration (Fig. 4a), which indicates that far fewer motors are initiated by the mismatch variants. Thus, the DNAzyme motor can effectively differentiate the fully matched target from these variants of single-base mismatch. The selection factor ranges from 5.1 to 16.7 depending on the mismatch site (Supplementary Table 2). The high specificity of the DNAzyme motor is attributed to the design of the locking strand that contains a sequestering domain in addition to the target-binding domain[40] (Supplementary Fig. 10).

Having optimized the operating conditions of the DNAzyme motor responding to the DNA target, we further examined the operation of the DNAzyme motor in response to various concentrations of the target miRNA. As expected, higher concentrations of the target miRNA led to larger fluorescence increases (Fig. 4b), consistent with more DNAzyme motors being initiated by the higher concentrations of the target miRNA. A linear relationship was observed between fluorescence intensity and target miRNA concentration from 5 to 200 pM (Supplementary Fig. 11). The operation of the motor initiated by 1 pM target can generate a fluorescence increase distinguishable from the blank, indicating the high sensitivity of the motor. The motor responds similarly to both the target miRNA (Fig. 4b and Supplementary Fig. 11) and DNA (Supplementary Fig. 12).

We have determined that on average ~12 locked DNAzyme motors were conjugated on each AuNP. With a concentration of 230 pM AuNPs used in the operation, the total number of the DNAzyme motors are in large excess over the target miRNA when the concentrations of miRNA are lower than those of AuNPs. Therefore, only one (or none) of the DNAzyme motors on each AuNP is activated by the target miRNA. We reason that these activated DNAzyme motors operate similarly and independently. To test this, we monitored the operating curves resulting from 50, 100 and 200 pM target miRNA (Supplementary Fig. 13). The operation of individual DNAzyme motors follows a similar

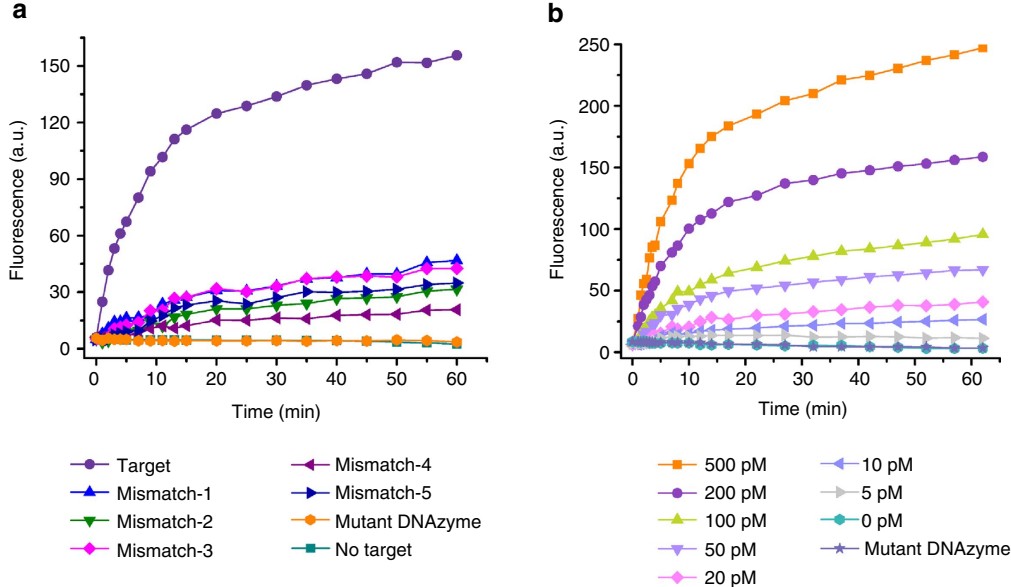

**Figure 4 | Operating curves showing specificity and sensitivity of the DNAzyme motor.** (**a**) Operating curves of the DNAzyme motor tested with the target DNA sequence and five variants of single-base mismatch. Sequences of the mismatch variants are shown in Supplementary Table 1. RSDs were 1.4–7.6%. (**b**) Operating curves of the DNAzyme motor in response to varying concentrations (0–500 pM) of the target miRNA. Different concentrations (0–500 pM) of the target miRNA were added into motor solutions each containing the equivalent of 230 pM AuNP. The solutions were incubated for 20 min to ensure the strand displacement reaction was completed. Then 500 µM $Mn^{2+}$ was added into the solutions to activate the DNAzyme. The time at which $Mn^{2+}$ was added is referred to as time 0. RSDs were 1.1–9.3%.

profile for three target miRNA concentrations. These results also indicate that a single DNAzyme motor walks about 30 steps in 30 min (Supplementary Fig. 13).

To trace the operation of the DNAzyme motor on individual AuNPs, we designed an alternative system that enables each walking step of the motor to turn on the fluorescence of a Cy5 molecule on the AuNP (Supplementary Fig. 14 and Supplementary Table 3). We designed the substrate strand to have a hairpin structure with a long single-stranded overhang that hybridizes to a Cy5-labelled DNA strand. The end of the substrate strand without the overhang is labelled with a black hole quencher so that hybridization of the substrate strand to the Cy5-labelled strand quenches the fluorescence of Cy5. We conjugated onto each AuNP dozens of locked DNAzyme strands and hundreds of Cy5-labelled strands to which the hairpin substrates are hybridized. In the presence of the target miRNA, the locked DNAzyme is activated to cleave the substrate at the single-ribonucleotide junction in the hairpin loop. The cleavage disrupts the hairpin structure and releases a quencher-containing segment from the AuNP, restoring the fluorescence of Cy5. The DNAzyme dissociates from the cleaved substrate and hybridizes to the next substrate, achieving the walking of the DNAzyme motor. Each walking step restores the fluorescence of a Cy5 molecule that remains attached to the AuNP. Our results (Supplementary Movie 1a and Supplementary Fig. 15a) indeed show that the fluorescence of Cy5 from individual AuNPs increases over time, representing the stepwise walking of the DNAzyme motor on the AuNP and the corresponding catalytic cleavage of the quencher-labeled substrate. In the absence of the target miRNA, there is very little fluorescence background (Supplementary Movie 1b and Supplementary Fig. 15b), consistent with the fact that the DNAzyme motor is inactive.

**Intracellular operation of the DNAzyme motor.** We further applied the motor to its operation in living cells (Fig. 5). We used

the MDA-MB-231 cell line, derived from human breast adeno-carcinoma cells, to test the intracellular operation of the DNAzyme motor. The target miRNA miR-10b is present in this cell line at a very low concentration and it would be difficult to image it using other methods[41–43]. We functionalized the DNAzyme motor and its track on AuNPs, which facilitates the cellular uptake of the motor system. Previous studies have demonstrated that DNA-functionalized AuNPs can be efficiently taken up by cells without the need for transfection reagents[44–46]. We determined the cellular uptake of the DNAzyme motor system by measuring the concentration of Au inside cells. The measured amount of Au, equivalent to the number of DNAzyme motors in the cells, increases with increase in concentration and time of incubation (Supplementary Fig. 16). When 0.2 nM AuNP and 2 h incubation were used, each cell took up about $3.2 \times 10^4$ AuNPs, which is approximately equivalent to 13 nM AuNP in the cell.

Inside the cells, the specific miR-10b miRNA hybridizes to the locking strand of the motor system through the strand displacement reaction, exposing the sequestered **Arm 2** of the DNAzyme motor and freeing the DNAzyme to interact with a substrate strand. Further treatment of the cells with 5 mM $MnCl_2$ initiated the DNAzyme motor to walk along the AuNP autonomously and processively. Each walking step restores the fluorescence of one FAM molecule. After 60 min operation, the fluorescence is detectable from the MDA-MB-231 cancer cells (Fig. 5d), suggesting that the intracellular operation of the DNAzyme motor has been accomplished. These cells show various fluorescence intensities, suggesting that the intracellular miR-10b levels are different among these cells. When cells were not treated with $Mn^{2+}$ solution, no fluorescence was observed (Fig. 5b), confirming that the operation of the DNAzyme motor requires both the target miRNA and the cofactor. This control experiment also suggests that the substrate strand on the AuNPs is stable and is not released without the active operation of the DNAzyme motor. We also tested the mutant DNAzyme motor

incubated with the target MDA-MB-231 cells. As expected, no fluorescence was observed (Fig. 5c), proving the high stability of the substrate strand on AuNPs in the absence of an active DNAzyme motor. Another control, using AuNPs conjugated with the substrate strands but not with the DNAzyme, showed no fluorescence (Fig. 5a), further proving that substrate strands on the AuNP are stable within the cells. We also introduced the motor system into two control cells deficient in miR-10b, MCF-10a and MCF-7 cells[41]. After 60 min incubation, little fluorescence is observed from these two cells (Fig. 5e,f), proving the specificity of the DNAzyme motor for the specific miRNA target.

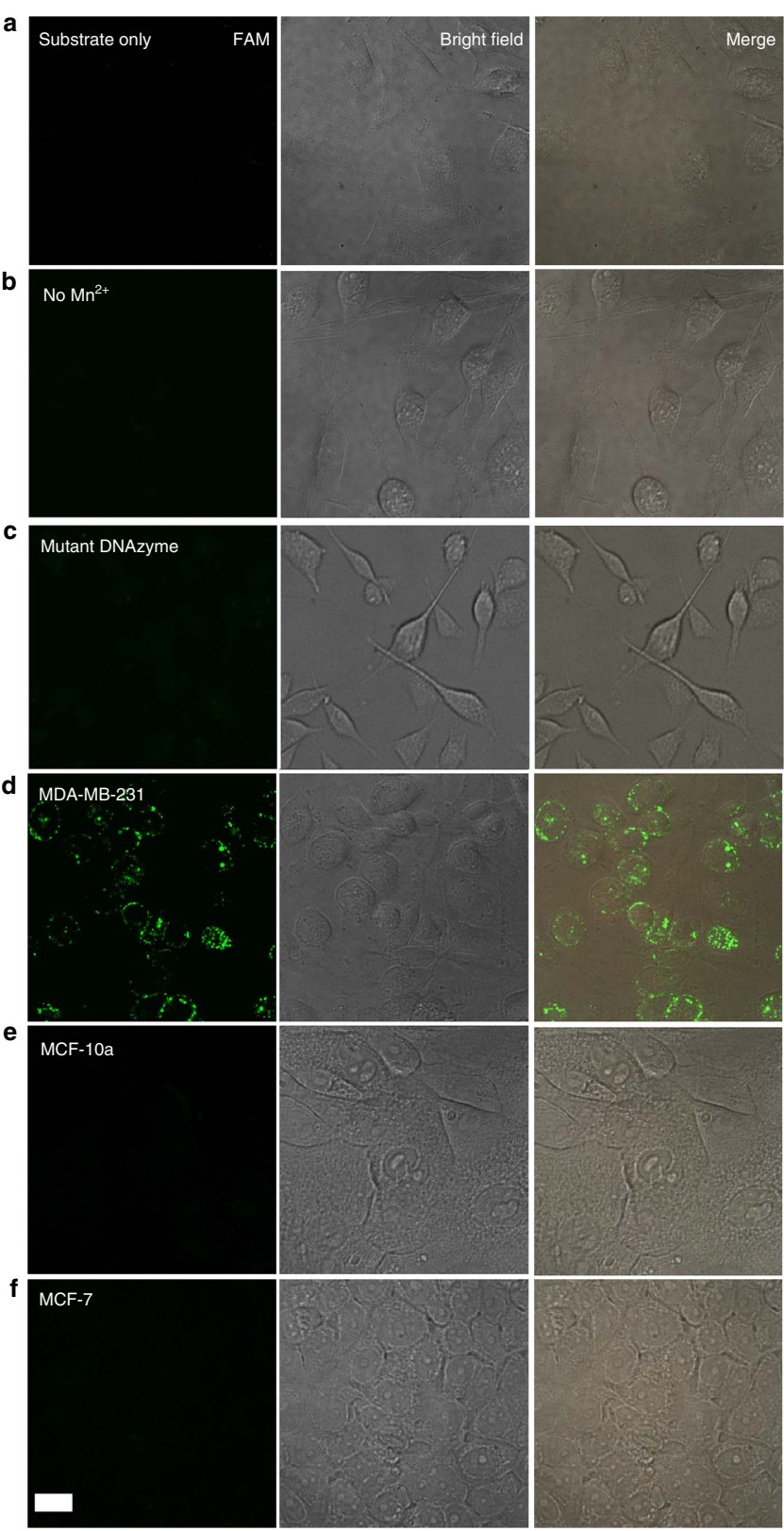

$Mn^{2+}$ is required for the operation of the DNAzyme motor. Although native cellular $Mn^{2+}$ levels are not sufficient to activate the DNAzyme motor, the levels of $Mn^{2+}$ required can be readily taken up by cells through simple incubation of the cells with $Mn^{2+}$ solution.[39] We tested the intracellular operation of the DNAzyme motor by treating cells with different concentrations of $Mn^{2+}$, 1, 5 and 10 mM. Similar fluorescence intensities were obtained, suggesting that the DNAzyme motor operates in a similar manner in these three cases (Supplementary Fig. 17). The reliable operation of the DNAzyme motor within a wide range of $Mn^{2+}$ concentrations makes intracellular operation of the motor practical.

To examine the autonomy and processivity of the motor walking, we imaged cellular fluorescence at different time points of the 60 min operation (Fig. 6). The fluorescence increases steadily for all cells over the operating time, indicating that the DNAzyme motor walks along AuNP autonomously and processively in living cells (Fig. 6, Supplementary Fig. 18 and Supplementary Movie 2). We further quantified the fluorescence increase in individual cells (Fig. 6c). The different slopes of the fluorescence increase imply that the target miRNA is present at different levels in these cells. This target-initiated operation of the motor can be used for in situ amplified detection of target miRNA in living cells. No fluorescence was observed from the target cells (Supplementary Fig. 19) when the DNAzyme motor system was constructed with the mutant DNAzyme.

We further detected the fluorescence of the target MDA-MB-231 cells after the cells were incubated with either the functional DNAzyme motor system (Supplementary Fig. 20a) or the mutant DNAzyme motor system (Supplementary Fig. 20b) for 1, 3 and 5 h. Cells treated with the functional DNAzyme motor system showed fluorescence throughout 5 h (Supplementary Fig. 20a). The slight decrease in fluorescence intensity over the longer time could be due to fluorescence bleaching. Non-detectable fluorescence from the cells treated with the mutant DNAzyme motor system throughout the 5 h period (Supplementary Fig. 20b) indicates negligible release of the fluorescent substrate from the AuNPs, that is, good stability of the DNAzyme motor system in the target cells.

In addition to amplified detection of the target miRNA in living cells, the DNAzyme motor system enables fluorescence determination of the cellular location of the target miRNA. The target miRNA displaces the DNAzyme strand to form a duplex with the locking strand (Supplementary Fig. 2). The fluorescence of Cy5 in the duplex is then restored and can be used to indicate the location of the target miRNA. We imaged the fluorescence of both Cy5 and FAM in the cells (Fig. 7). The fluorescence intensity from the Cy5 image is much weaker than the fluorescence intensity from the FAM image at the end of 60 min operation. This is consistent with the fact that the DNAzyme motor amplifies the signals for imaging, because the DNAzyme motor generates multiple FAM-labelled **F1** molecules ($\sim 30$) in response to a single target miRNA strand.

We also determined the relative amount of substrate strands that were cleaved in response to the DNAzyme motor operation (Fig. 8). For comparison, we subsequently treated cells with 10 mM 2-mercaptoethanol to release all remaining substrate strands from AuNPs and then imaged the total FAM fluorescence. These results indicate that on average 22% of the total substrate strands on the AuNP are cleaved as a result of the DNAzyme motor action.

## Discussion

We demonstrate, for the first time, the accomplishment of the operation of a synthetic DNA motor in living cells. This DNAzyme motor system has several important features desirable for intracellular operation and characterization. First, the entire motor system is a functionalized AuNP that is decorated with both the DNA motor and its track. The integration of the motor and its track on a single AuNP facilitates the cellular uptake of the motor system. DNA-functionalized AuNPs can be efficiently taken up by cells without the need for transfection reagents that would usually be required for cellular uptake of DNA strands[47]. In addition, previous studies have shown that AuNPs at similar concentration as the one we used had little effect on the cytotoxicity and viability of cells[48,49]. Second, the motor is self-powered, enabling autonomous motion without the need to add fuel DNA strands and/or protein enzymes. This is of great importance for intracellular operation, because the external addition of fuel DNA strands and/or protein enzymes is not practical. Third, the intracellular operation of the motor is initiated by the specific cellular target, for example, miRNA. Outside the cells, the DNAzyme motor is silenced by a purposely designed locking strand. However, once inside the cells, the specific cellular miRNA hybridizes to the locking strand, initiating the operation of the motor. The motor system is highly specific, enabling differentiation of the fully matching target from various sequences containing a single-base mismatch at different sites. Fourth, the motor operates reliably under physiological pH conditions and reasonable cofactor concentrations, which enables meaningful applications to live cells. Furthermore, 30 walking steps of the motor can be accomplished within 30 min. This high walking speed of the motor is achieved by the rational control of the arm length of the DNAzyme and by the construction of high-density tracks on AuNP. Finally, the result of the intracellular operation of the motor can be monitored in real time by using fluorescence imaging. Each walking step of the motor restores the fluorescence of a previously quenched FAM molecule, enabling real-time imaging of the progression of the motor. Importantly, the motor system enables amplified detection of specific miRNA in living cells. The operation of the motor can release many FAM-containing **F1** strands from AuNPs in response to a single miRNA target. Furthermore, labelling of the locking strand with a second

**Figure 5 | Imaging of live cells after uptake of the DNAzyme motor system.** (**a**) Images showing MDA-MB-231 cells after incubation with AuNPs functionalized with the substrate strand but not the DNAzyme strand (negative control). (**b**) Images showing MDA-MB-231 cells after incubation with the DNAzyme motor system but not subsequently treated with the cofactor $Mn^{2+}$ (negative control). (**c**) Images showing the target MDA-MB-231 cancer cells after incubation with a mutant DNAzyme motor system and the subsequent treatment with the cofactor $Mn^{2+}$ (negative control). (**d**) Images showing the target MDA-MB-231 cancer cells after incubation with the DNAzyme motor system and the subsequent treatment with the cofactor $Mn^{2+}$. Fluorescence images of the target cells are the result of intracellular operation of the DNAzyme motor. (**e**) Images showing the negative control MCF-10a cells after the same treatment as for the MDA-MB-231 cells in **d**. (**f**) Images showing the negative control MCF-7 cells after the same treatment as for the MDA-MB-231 cells in **d**. All images were collected at the end of 60 min operation time (i.e., 60 min after the addition of the cofactor $Mn^{2+}$). Fluorescence imaging of living cells was performed using an Olympus IX-81 microscope equipped with a Yokogawa CSU $\times 1$ spinning disk confocal scan-head and a Hamamatsu EMCCD camera with $\times 40/1.3$ Oil and $\times 20/0.85$ Oil objective lenses. A pumped diode laser at 491 nm was used for the excitation of FAM. Magnification was chosen to allow a final pixel size of 447 nm with $\times 20$ lens and 223 nm with $\times 40$ lens. The length of the scale bar is 17 μm.

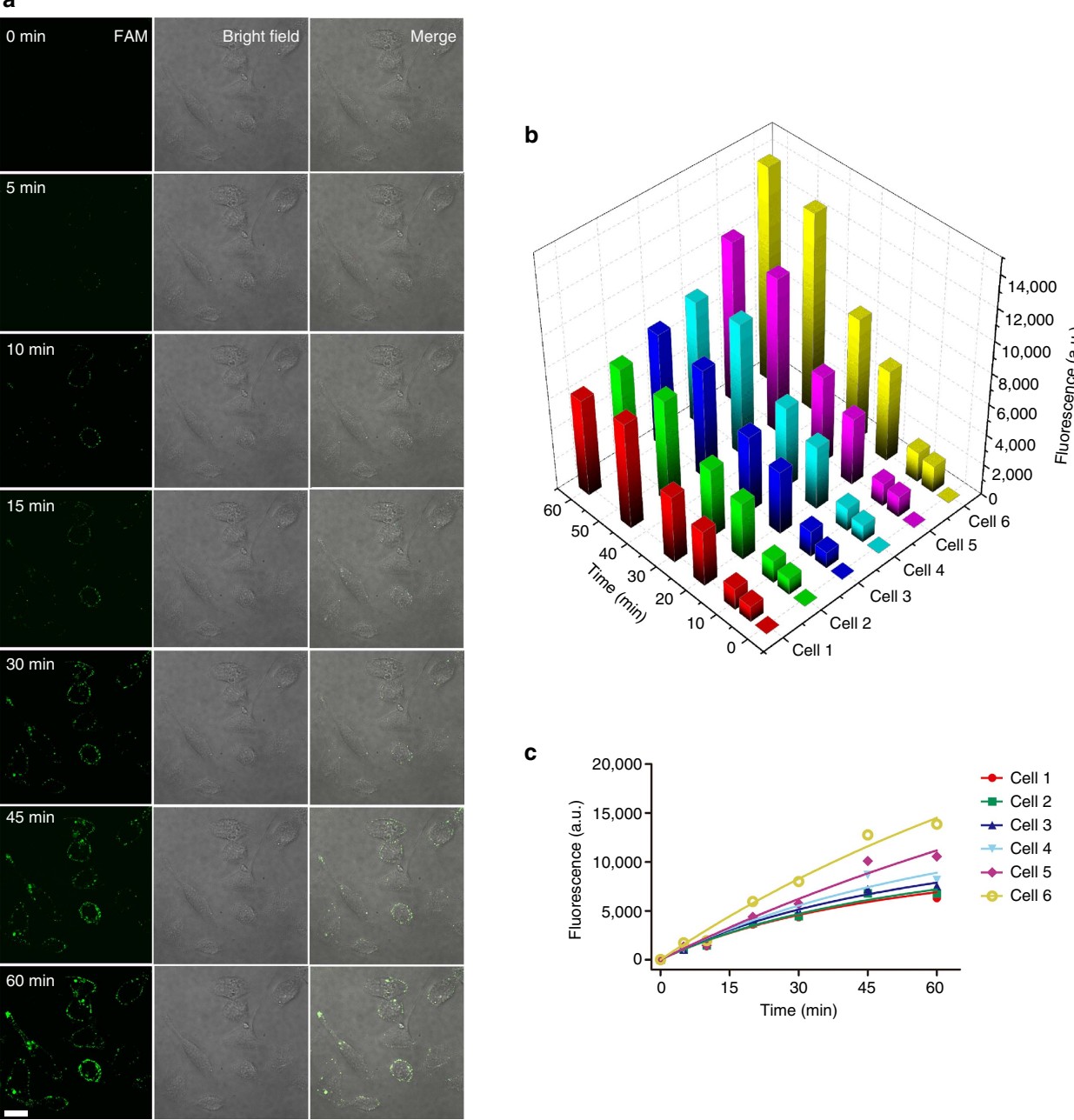

**Figure 6 | Images and fluorescence intensity of target cells upon operation of the DNAzyme motor.** The DNAzyme motor was designed to respond to the intracellular miRNA miR-10b. (**a**) Images of MDA-MB-231 cancer cells after intracellular operation of the DNAzyme motor for 0, 5, 10, 15, 30, 45 and 60 min. 0 min refers to the time point when $Mn^{2+}$ is added into the operating buffer. The LMM5 laser transmission setting was 85 and the laser excitation time was 185 ms for each image. (**b**,**c**) Fluorescence intensity of six cells over the 60 min operation time of the DNAzyme motor. The fluorescence intensity was measured using ImageJ 1.47. The length of the scale bar is 17 μm.

fluorophore, Cy5, allows the motor system to signal the location of the target miRNA.

Each walking step involves three actions, hybridization of the DNAzyme to one substrate strand to form a DNAzyme–substrate complex, cleavage of the substrate strand to create a DNAzyme–product complex and release of the DNAzyme from cleavage products to regenerate the free DNAzyme[50]. The cleavage of the substrate is not the rate-determining step, because the single-turnover cleavage rate ($4.7\,min^{-1}$) is higher than the multiple-turnover cleavage rate ($3.1\,min^{-1}$) of the first 5 min operation and much higher than the cleavage rate of the

subsequent operation time. The cleavage products, **F1** and **F2**, are constant for all walking steps, thus the release rate of DNAzyme from cleavage products is also constant. However, the walking speed of the motor varies over time, faster in the first 10 min and slower afterwards. This results mainly from the varying hybridization rate of the DNAzyme to the substrate strand. In the initial stage of walking, many substrate strands are near the conjugation site of the DNAzyme strand. Therefore, hybridization of the DNAzyme to the substrate strands is fast, leading to a high walking speed. As the nearby substrates are cleaved, the hybridization of the DNAzyme to

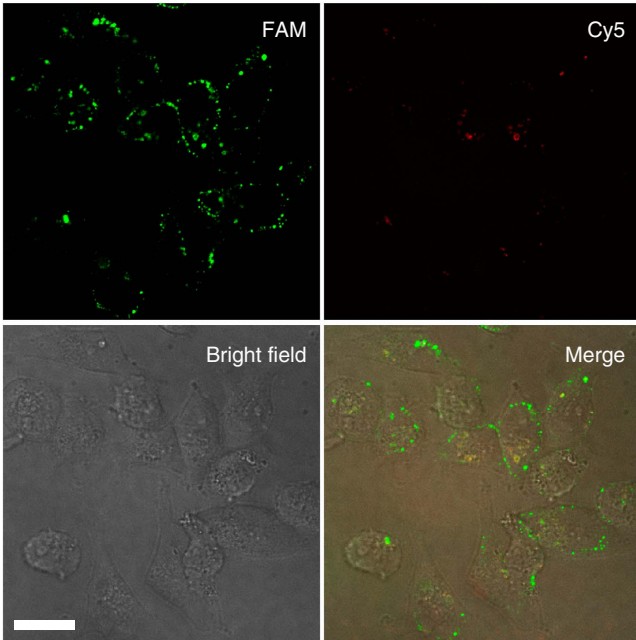

**Figure 7 | Images of MDA-MB-231 cancer cells from the fluorescence detection of FAM and Cy5.** The higher intensity from FAM is due to the intracellular operation of the DNAzyme motor, with each walking step restoring fluorescence of a FAM molecule. Therefore, the DNAzyme motor results in the amplified imaging of the intracellular miRNA target. The Cy5 image is much weaker because there is no amplification for the Cy5 signal. Fluorescence imaging of living cells was performed using an Olympus IX-81 microscope equipped with a Yokagawa CSU × 1 spinning disk confocal scan-head and a Hamamatsu EMCCD camera with × 40/1.3 Oil and × 20/0.85 Oil objective lenses. Two pumped diode lasers at 491 nm and 630 nm were used for the excitation of FAM and Cy5. Magnification was chosen to allow a final pixel size of 447 nm with × 20 lens and 223 nm with × 40 lens. The length of the scale bar is 17 μm.

the substrate becomes slower, because the hybridization requires stretching of the spacer **S2** to access the distant substrate strands. Finally, no substrate strand is accessible to the DNAzyme and the walking of the motor stops. We compared the walking of the miRNA-initiated DNAzyme motor and a control DNAzyme motor that is not conjugated to AuNP (Supplementary Fig. 21). The miRNA-initiated motor had a faster initial response to the target (Supplementary Fig. 21c) and then slowed and finally stopped after 3 h. The free control DNAzyme had a constant speed over the 6 h operation (Supplementary Fig. 21c), because in the absence of the spacer **S2**, the hybridization rate of the control motor to the substrate remains constant. The initial walking speed of the miRNA-initiated motor is significantly higher than that of the control motor, because the walking of the control motor from the cleaved substrate to the next substrate involves a strand displacement reaction, slowing the walking speed.

The response of the DNAzyme motor is not limited to the miR-10b miRNA. Similar motor systems can be readily constructed to respond to other miRNA and messenger RNA targets. A modification to the design can be made by simply altering the target binding domain of the locking strand. Diverse DNAzyme motors can also be designed to respond to small molecules and proteins in cells. In addition, by incorporating functional molecules (for example, therapeutic molecules and antisense strands) into the substrate strands, the DNAzyme motor system can be used for target-triggered drug release and modulation of cellular activity. Therefore, we envision many applications of the DNAzyme motor strategy, such as sensing intracellular molecules, imaging live cells, regulating cellular functions and facilitating drug delivery.

## Methods

**Construction of the miRNA-initiated DNAzyme motor system.** The DNAzyme motors were constructed on 20-nm AuNPs by functionalizing the AuNPs with the pre-locked DNAzyme and its substrate. The sequences of the DNAzyme, substrate and locking strand are summarized in Supplementary Table 1, with complementary sequences shown in identical matching colors. The DNAzyme

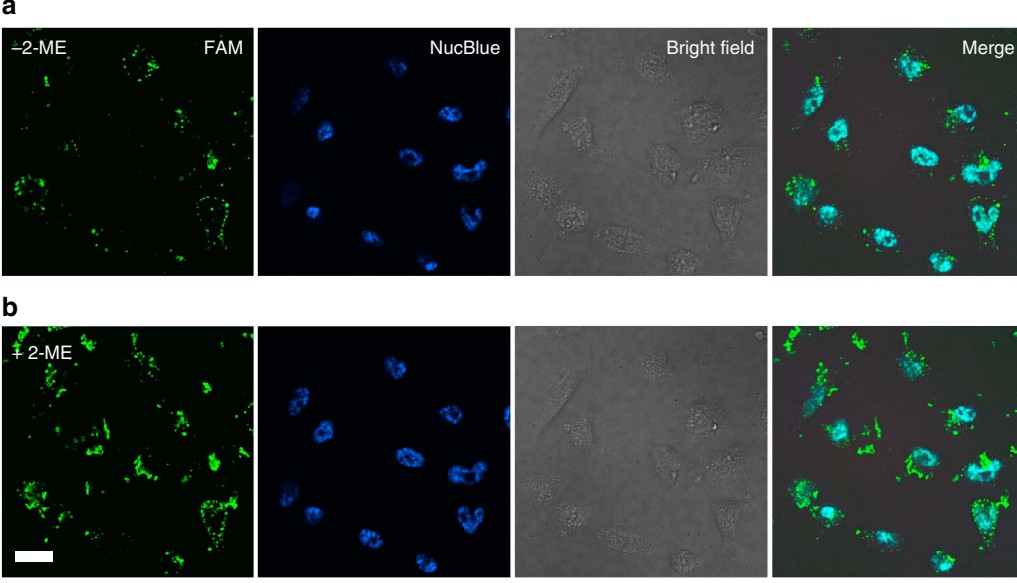

**Figure 8 | Images of cells after intracellular operation of the DNAzyme motor and chemical cleavage.** (**a**) Images of MDA-MB-231 cancer cells after intracellular operation of the DNAzyme motor for 60 min. (**b**) Images of MDA-MB-231 cancer cells after additional treatment with 10 mM 2-mercaptoethanol (2-ME). 4′,6-diamidino-2-phenylindole (DAPI) was used to stain the nucleus of the cells. Fluorescence imaging of living cells was performed using an Olympus IX-81 microscope equipped with a Yokagawa CSU × 1 spinning disk confocal scan-head and a Hamamatsu EMCCD camera with × 40/1.3 Oil and × 20/0.85 Oil objective lenses. Two pumped diode lasers at 405 and 491 nm were used for the excitation of DAPI and FAM. Magnification was chosen to allow a final pixel size of 447 nm with × 20 lens and 223 nm with × 40 lens. The length of the scale bar is 17 μm.

strand and the substrate strand that require direct conjugation to the AuNPs were thiolated. Before conjugation to the AuNPs, the DNAzyme was locked (silenced) by using a locking strand. For the preparation of the locked DNAzyme strand, the locking strand and the DNAzyme strand at a molar ratio of 3:1 were mixed in $1 \times$ PBS buffer (pH 7.4). The use of three-fold molar excess of the locking strand was to ensure the complete locking of the DNAzyme strand by the locking strand. The mixture was heated to 75 °C and gradually cooled to 4 °C at a rate of 1.2 °C min$^{-1}$. The locked DNAzyme strand and the substrate strand were then conjugated to the AuNPs. AuNPs (20 nm diameter), the locked DNAzyme strand and the FAM-labelled substrate strand were mixed at a molar ratio of 1:50:1,000, to control the ratio of the DNAzyme and the substrate on the AuNPs. This solution was incubated at room temperature for 12 h. Tween 20 (1%) was then added to make the final solution contain 0.05% Tween 20, to reduce adsorption and aggregation of AuNPs. To enhance the DNA loading amounts, NaCl was added in increments of 0.05 M for the first two times and thereafter in increments of 0.1 M for six more times. After each addition of NaCl, the solution was sonicated for 1 min followed by incubation for 40 min at room temperature. After incubation at room temperature for an additional 24 h, the solution was centrifuged at 16,000 g for 20 min to separate the AuNPs from the unconjugated DNA. The AuNPs were washed four times using 1 ml of 10 mM Tris-HCl pH 7.4 containing 0.05% Tween 20. The AuNPs were resuspended in 25 mM Tris-HCl pH 7.4 at a concentration of 2.3 nM and stored at 4 °C until use.

**Determination of the number of substrate molecules per AuNP.** To determine the average number of the substrate molecules on each AuNP, the conjugated substrate strands were first released from the AuNPs using 2-mercaptoethanol. The solution was then centrifuged to precipitate the AuNPs and the supernatant containing the released substrate strands was measured by fluorescence. Specifically, 10 µl of 2.3 nM AuNP solution was mixed with 10 µl of 35 mM 2-mercaptoethanol and the mixture was then diluted to 100 µl by using $1 \times$ PBS buffer. The mixture was placed in the dark. After an overnight incubation at room temperature, the solution was centrifuged at 16,000 g for 10 min, to precipitate the AuNPs. A 95 µl supernatant was transferred onto a 96-well plate (Fisher Scientific, Ottawa, ON), which was then loaded onto a fluorescence microplate reader (Beckman Coulter, DTX 800) for fluorescence detection. Molar concentrations of the substrate were determined against a calibration of the FAM-labeled substrate strand. The average number of substrates per AuNP was then derived from the concentrations of AuNPs and substrate. Our results show that on average, approximately 232 substrate molecules were conjugated to each AuNP. On the basis of a molar ratio of 1:20 for the DNAzyme strand and the substrate strand used together in the conjugation reaction, ∼12 DNAzyme molecules were conjugated to each AuNP. Thus, the densities of the substrate and DNAzyme strands are $1.8 \times 10^{-1}$ and $9.2 \times 10^{-3}$ nm$^{-2}$ on the 20 nm AuNPs.

**Examination of locking efficiency of the DNAzyme strand.** The locking efficiency of the DNAzyme strand by the locking strand was examined by using native PAGE (Supplementary Fig. 4). Three microlitres of 10 µM DNAzyme strand and 9 µl of 10 µM locking strand were mixed and 18 µl of $1 \times$ PBS was then added to make a total volume of 30 µl. The mixture was then heated to 75 °C by using a Bio-Rad thermal cycler (Bio-Rad, Hercules, CA) and cooled down to 4 °C at a rate of 1.2 °C min$^{-1}$. PAGE on 10% polyacrylamide gel was used to separate the locked DNAzyme strand from the excess locking strand. Electrophoresis was carried out at 80 V for 90 min. After separation, the gel was stained with ethidium bromide for 30 min and imaged by a fluorescence gel imaging system (ImageQuant LAS 4000, GE Healthcare Life Sciences, Pittsburgh, PA).

**Examination of displacement of target from locked DNAzyme.** The strand displacement efficiency of the DNAzyme strand by the target sequence was examined by using native PAGE. Three microliters of 10 µM DNAzyme strand and 3 µl of 10 µM locking strand were mixed and annealed by using the temperature programme as described above. After the DNAzyme strand was bound by the locking strand, 2.7 µl of 10 µM target sequence was added and $1 \times$ PBS buffer was used to make a final volume of 30 µl. After incubation at room temperature for 30 min to ensure the completion of the strand displacement reaction, 5 µl of the solution was mixed with 5 µl 2 × loading buffer and then loaded onto a 10% polyacrylamide gel for separation.

**Determination of single-turnover cleavage rate of DNAzyme.** To determine the single-turnover cleavage rate of the DNAzyme (Supplementary Fig. 6), the full-length DNAzyme (containing an 8 nt **Arm 1** and an 8 nt **Arm 2**) was used to ensure the formation of a stable DNAzyme–substrate complex. One hundred and ninety microlitres of a mixture was prepared to contain the DNAzyme and FAM-labeled substrate strand at a molar ratio of 10:1 in 25 mM Tris-acetate (8.0) and 200 mM NaCl. The use of ten-fold molar excess DNAzyme was to ensure the complete hybridization of the substrate strand to the DNAzyme. After incubation for 10 min, 10 µl of 200 mM Mn$^{2+}$ or 10 mM Mn$^{2+}$ was added to initiate the DNAzyme-catalysed cleavage. At designated time points, 10 µl aliquots of reaction solution were sampled and quenched by using 10 µl of 50 mM EDTA and 8 M urea. Reaction solutions were loaded onto a 14% denaturing polyacrylamide gel for separation. Electrophoresis was carried out at 80 V for 80 min in a water bath set at 50 °C.

**Comparison of DNAzymes for construction of the DNAzyme motor.** The selection of the DNAzyme to construct the motor is of fundamental importance for accomplishing the intracellular motion of the motor. DNAzymes with high rate of catalytic cleavage, short catalytic core sequence and editable arm sequences are preferred. The high rate of catalytic cleavage can allow the motor to have a fast walking speed. The short catalytic core provides options for the DNAzyme to have longer spacers that may be needed for the walking of the motor. The editable arms make it possible to truncate the arms of the DNAzyme for the motor construction. We compared features of three DNAzyme candidates, 10–23, 8–17 and 8–17E. The 10–23 and 8–17 DNAzymes require high concentrations (>50 mM) of Mg$^{2+}$ to achieve their optimal activity. It is challenging for cells to take up such high levels of Mg$^{2+}$ ion[51,52]. We chose 8–17E, a variant of the 8–17 DNAzyme, to construct the DNAzyme motor, because previous work has shown that 8–17E DNAzyme could reach its best activity in the presence of 200 µM Pb$^{2+}$. We found that truncation of **Arm 1** and **Arm 2** of the original 8–17E DNAzyme from 9 to 5 and 7 nt alters the dependence of the DNAzyme on divalent metal ions. Instead of Pb$^{2+}$, Mn$^{2+}$ results in the highest activity of this DNAzyme and the optimal concentration of Mn$^{2+}$ is 500 µM. This alteration is much more favourable for intracellular operation, because Mn$^{2+}$ is much less cytotoxic than Pb$^{2+}$ and cells can quickly and readily take up the amount of Mn$^{2+}$ required for the operation of the motor. We also compared the operation of DNAzyme motors constructed from the use of 10–23, 8–17 and 8–17E DNAzymes. Although the binding arms are the same in these three DNAzyme motors, in the presence of 500 µM Mn$^{2+}$, the DNAzyme motor constructed from 8–17E showed a significantly higher walking speed than the other two motors (Supplementary Fig. 22).

**Evaluation of the operation of the DNAzyme motor in buffer.** A DNA strand with the same sequence as miR-10b miRNA was used as the initial target to turn on the operation of the DNAzyme motor. The impact of key parameters on the operation of the DNAzyme motor was examined, including locking strands, cofactors, operating pH, and DNAzymes. Unless otherwise stated, 200 pM target sequence and 230 pM functionalized AuNPs were used to evaluate the operation of the DNAzyme motor in buffer. Ninety-five microlitres of the operating solutions were prepared to contain 200 pM target sequence (or no target sequence in parallel experiments to serve as reagent blanks) and 230 pM functionalized AuNP in 25 mM Tris-acetate buffer (pH 8.0) and 200 mM NaCl. After incubation at room temperature for 20 min, a Mn$^{2+}$ solution (5 µl, 10 mM) was added to initiate the operation of the motor. Fluorescence was measured at 535 nm in real time for 60 min with excitation at 485 nm.

The response of the motor to varying concentrations of the target sequence was evaluated under the optimized conditions. Ninety-five microlitres of the operating solutions were prepared to contain 230 pM functionalized AuNP and varying concentrations of the target sequence in 25 mM Tris-acetate buffer (pH 8.0) and 200 mM NaCl. After incubation at room temperature for 20 min, 5 µl of 10 mM MnCl$_2$ solution was added to initiate the operation of the motor. The fluorescence of the solutions was then measured in real time with excitation at 485 nm and emission at 535 nm.

**Examination of cellular uptake of the motor system.** The cellular uptake of the DNAzyme motor system, consisting of AuNPs functionalized with substrate and locked DNAzyme sequences, was determined by using inductively coupled plasma mass spectrometry (Supplementary Fig. 23). Cells were seeded onto an 18-mm round glass slide. When cultured to 80–90% confluence, cells were washed with $1 \times$ PBS three times. To the glass slide was added 100 µl of the uptake medium prepared by suspending different concentrations of the DNAzyme motor system into Opti-MEM Reduced Serum Medium (Fisher Scientific). After incubation at room temperature for 2, 6 or 8 h, to allow cells to take up the DNAzyme motor system, cells were thoroughly washed with $1 \times$ PBS six times. Cells were then detached by using 0.05% trypsin-EDTA and collected using centrifugation. The number of cells was counted by using a haemocytometer. Collected cells were lysed and digested with 10% ultrapure nitric acid at 60 °C overnight. The amount of AuNPs was measured by detecting Au at $m/z$ 197 using inductively coupled plasma mass spectrometry (Agilent 7500cs, Japan), against a calibration of acid-digested AuNP standards. The uptake number of AuNPs per cell was then derived from the total amount of AuNPs and total cell numbers. Based on the results of cell imaging, the cell size of 20 µm diameter was used to estimate the intracellular AuNP concentrations.

**Evaluation of intracellular operation of the DNAzyme motor.** All the cell lines were cultured in a humidified incubator at 37 °C containing 5% CO$_2$. The MCF10a, MCF-7 and MDA-MB-231 cell lines were obtained from the American Type Culture Collection (ATCC, Manassas, VA). The MCF10a was cultured in DMEM/F12 medium (Invitrogen Life Technologies, Carlsbad, CA) supplemented with 20 ng ml$^{-1}$ epidermal growth factor, 100 ng ml$^{-1}$ cholera toxin, 500 ng ml$^{-1}$ hydrocortisone, 2 mM L-glutamine and 20 ng ml$^{-1}$ gentamicin. The MCF-7 cell line was cultured in DMEM medium, supplemented with 10% fetal bovine serum,

penicillin and streptomycin, and 2.5 mM L-glutamine (GIBCO-Invitrogen, Carlsbad, CA). The MDA-MB-231 cell line was cultured in RPMI-1640, supplemented with 10% fetal bovine serum, penicillin and streptomycin, and 2.5 mM L-glutamine (GIBCO-Invitrogen).

Cells were seeded onto an 18 mm round glass slide. When cultured to 80–90% confluence, cells were washed with $1 \times$ PBS twice and incubated with Opti-MEM Reduced Serum Medium containing the DNAzyme motor (equivalent to 0.2 nM functionalized AuNP) for 2 h, to allow the cellular uptake of the DNAzyme motor. To remove the DNAzyme motor not taken up, cells were washed with $1 \times$ PBS three times and with 25 mM Tris-acetate buffer (pH 8.0) containing 125 mM NaCl another three times. Cells were treated with the 25 mM Tris-acetate buffer (pH 8.0) containing 125 mM NaCl and 5 mM $MnCl_2$, to allow the cellular uptake of $Mn^{2+}$. Fluorescence imaging of living cells was carried out on an Olympus IX-81 microscope that was coupled with a Yokagawa CSU × 1 spinning disk confocal scan-head and a Hamamatsu EMCCD camera with × 40/1.3 Oil and × 20/0.85 Oil objective lenses. Two pumped diode lasers at 491 and 630 nm were used for the excitation of FAM and Cy5, respectively. The exposure time was set to be 185 ms for FAM and 100 ms for Cy5.

To test whether possible adsorption of the DNAzyme motor on the cell surface could produce fluorescence signals that would confound the detection of intracellular targets, we conducted the following control experiment (as schematically depicted in Supplementary Fig. 24). We first incubated MDA-MB-231 cancer cells with an inactive DNAzyme motor system constructed with the mutant DNAzyme. We then added 0.5 mM $Mn^{2+}$, incubated for 60 min and obtained fluorescence images of the cells. No fluorescence signal was detected (Supplementary Fig. 25b), which is as expected because the mutant DNAzyme is not able to cleave the substrate. After washing the cells three times with $1 \times$ PBS, we then incubated the cells with 200 pM free control DNAzyme sequence and then measured the cellular fluorescence after incubation for an additional 20 min. As the free control DNAzyme has 8 nt in both **Arm 1** and **Arm 2** complementary to its substrate strand (Supplementary Table 1), it can readily hybridize to and cleave the substrate strand. Therefore, if the DNAzyme motor system were adsorbed on the surface of the cells, the free control DNAzyme would hybridize to the substrate strands and cleave them off from the DNAzyme motor system, generating a fluorescence signal. Our results showed no detectable fluorescence (Supplementary Fig. 25c), ruling out possible interference from potential adsorption of the DNAzyme motor on the cell surface. As a confirmation that the mutant DNAzyme motor system entered the cells, we finally treated the cells with 2-mercaptoethanol to release all fluorescent substrate from the AuNPs of the mutant DNAzyme motor and monitored the fluorescence in the cells (Supplementary Fig. 25d). The detectable fluorescence signals in the cells confirm the presence of the DNAzyme motor system. The fluorescence signals are due to the chemical cleavage of the Au-S linkage, releasing the fluorescent substrate from the AuNP. These results confirm that DNAzyme motors were taken up by the cells and that nonspecific interaction of the DNAzyme motor system with the cells did not confound the detection of specific miRNA target.

To examine whether the target miRNA could leak out of the cells and then initiate operation of the DNAzyme motor outside of the cells, we conducted the following control experiment. We added a DNAzyme reaction buffer, containing 25 mM Tris-acetate pH 8.0 and 125 mM NaCl, to the MDA-MB-231 cells, and removed the buffer either 1 h after incubation with the cells or immediately after its contact with the cells (1 min). We then added 230 pM of the DNAzyme motor and 0.5 mM $MnCl_2$ to these reaction buffer solutions and monitored the fluorescence (the operation of the DNAzyme motor) for 60 min (Supplementary Fig. 26). If the target miRNA had leaked out of the cells, they would have initiated operation of the DNAzyme motor and generated fluorescence signals. However, there was no detectable fluorescence increase (Supplementary Fig. 26), suggesting that leaking of the target miRNA from the cells into the reaction buffer was negligible. As a positive control, further addition of 200 pM target miRNA to the solution resulted in an expected fluorescence increase (Supplementary Fig. 26).

**Data availability.** Data supporting the findings of this study are available within the article and its Supplementary Information files and from the corresponding authors upon reasonable request.

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

## Acknowledgements

This study was supported by the Canada Research Chairs Program, the Canadian Institutes of Health Research (CIHR), the Natural Sciences and Engineering Research Council (NSERC) of Canada, Alberta Health and Alberta Innovates. We thank Katerina Carastathis for her help with editing the manuscript.

## Author contributions

H.P. and H.Z. performed the experiments. H.P., H.Z. and X.C.L. conceived the concept. H.Z., X.-F.L. and X.C.L. supervised the project. H.P., X.-F.L., H.Z. and X.C.L. wrote the manuscript.

## Additional information

**Competing financial interests:** The authors declare no competing financial interests.

