## [Peer Review File · Nature Communications]

Reviewers' Comments:

Reviewer #1 (Remarks to the Author)

This paper reports the operation of an artificial nanomotor system inside living cells. The motor system is activated in the cells by a miRNA target together with a cofactor (Mn^{2+}). The subsequent motor operation releases many copies of quenched 'signal' strands so that the amplified fluorescence serves as an effective indicator of the presence of the target miRNA. Artificial nanomotors are believed to have many potential applications inside living cells, but progress in this direction has been hard to come by. This study, though still a demonstration of principle, represents an important step forward and will likely inspire more to follow in future. I think that this paper is worth publication in Nature Communications after some issues are addressed or clarified (see below).

(1) The paper show enough experiments to prove the motor's in vitro operation (e.g., effectiveness of oligonucleotide target/cofactor activating the motor, fluorescence amplification, etc.). But evidence for the motor's in vivo operation is not as strong. For example, the FAM light spots form rings around a cell. This pattern is seen throughout Figs. 5-8 and especially clear in Fig. 6. It raises a question whether the light spots are produced by the Au-motor systems that remain outside the cells and are stuck on their surface. By simple logic, the light spots produced from an intracellular cause should spread all over a cell instead of forming a ring along the cell's edge. One more control experiment is necessary to rule out this possibility.

(2) Following (1), the paper briefly mentions an ICP-MS experiment to measure Au concentration inside the cells (Fig. S9). Is the experiment done using the Au particle with the DNA motors or without? How does the method distinguish between the Au-motor systems inside the cells from those stuck on the surface (outside)? Judging simply from the optical images, even some light spots are seemingly inside a cell, but they can be on top of the cell (outside it).

(3) A counterargument to (1) is perhaps that the miRNA trigger is present only inside the living cell. Any possibility of the miRNA leaking from the cells? What's the miRNA concentration in the cell, roughly?

(4) Accepting that the Au-motor systems are inside, a question still remains whether the effect is triggered by a single species as the miRNA or other factors. Can the intracellular presence of a target DNA be ruled out? Most of in vitro tests are done using the target DNA instead of the miRNA (miR-10b). I wonder, why don't use the miRNA throughout the in vitro and in vivo experiments?

(5) What's the zero of the operating time? Is it the time to add Mn^{2+} ? There seems to be no control over the timing of miRNA inside the cells.

(6) How are the motor's steps determined (Fig. S7)?

(7) 'Amplified imaging of the intracellular microRNA target' appears not accurate. It's not a real imaging since the amplified FAM emission signals only the presence of the miRNA, but not its whereabouts. The miRNA location is still from Cy5 emission, which is independent of the motor-induced FAM amplification.

Reviewer #2 (Remarks to the Author)

In this manuscript, the authors described strategy of constructing a DNAzyme motor system on a

gold nanoparticle (AuNP), which has four features. Integration of all motor components onto a single AuNP facilitates cellular uptake of the DNA motor system. They found that this system could allow real-time imaging of the intracellular operation of the motor.

The work is interesting but the following concerns need to be addressed.

1. The description of the concept of DNAzyme motor system is too brief. I am not sure if any random non-oriented walking without tracking could be defined as "motor". Also it is better if the authors could provide a tracking video for the fluorescent molecules releasing from AuNPs.
2. In figure 5a, Images of live cells after uptake of a DNAzyme motor system were provided. I am wondering what will be like when you extend the reaction time from 1h to 3h, 5h. Is it possible that after longer time incubation, the fluorescent molecules will be released from the AuNPs?
3. In figure S12, why was the fluorescence decreased for the miRNA initiated DNAzyme motor after 3 h. The authors claimed it is because of the photo bleaching, but did the control DNAzyme motor system undergo the same condition of photo bleaching? Could you please explain it more?
4. Except monitoring of the fluorescent molecules detached from the AuNP in real-time imaging, could you also track its walking traces? Because even without the release of fluorescence dye, the AuNPs were continually randomly diffusing from here to there. Also, was its walking orientated and rational designed?
5. Could you provide the probe density of both substrate strand and DNAzyme strand on AuNP?
6. How did authors define the moving steps in Figure S4 and Figure S7? What is the relationship between each moving step and fluorescence increasing?
7. Can you track each fluorescent molecule cleaved by DNAzyme which might indicate the exact position of your AuNP, even so, it is also difficult to distinguish which dye molecule belongs to which AuNP. The increase value of fluorescence intensity is a collective behavior.
8. Why each AuNP contained only a single activated DNAzyme motor when the target sequence concentration was less than 230 pM? How did you quantify the DNAzyme motors in the cells?

Reviewer #3 (Remarks to the Author)

This manuscript reports the design of gold nanoparticle/RNA cleaving DNAzyme based nanomachine for the detection of microRNAs in cells. Such a system has not been reported before and the approach is definitely novel and could open up potential applications for cellular imaging of microRNA and other targets. Overall the study was reasonably well designed and executed, and the manuscript is well written and easy to follow. However, I do have several concerns, which need to be addressed before I can recommend its publication in Nature Communications.

- 1) My biggest concern is: Is the DNAzyme actually driving the operation of the nanomachine? Most of the data presented seem to argue for this key claim of the work, but I am not totally convinced because there is a great possibility that RNases might have actually mediated the RNA cleavage reaction. RNases are ubiquitous and potent enzymes. Many research agents (buffers, salts, water) are contaminated with RNases and cells contain all kinds of RNases. Therefore, I am concerned about the validity of the claim. An important experiment that the authors must perform is the use of a mutant DNAzyme where the catalytic core of the DNAzyme is mutated from TCCGAGCCGGTCGAA to TCCGATCCGGTcTAA. This mutant DNAzyme should be used as a control to perform the key experiments throughout the study (Figures 2B, 3A, 4, 5, 6 and 7). Without these experiments, the validity of the claims is questionable.
- 2) Only the data in Figure 3B has error bars. Were replicates done for all the experiments? If so, the information should be provided and statistic analysis performed.
- 3) PAGE analysis for all the in vitro cleavage reactions (Figure 2, b, c, d; 3b, 4a and 4b) should be performed and the data should be provided in SI.

4) The authors need to comment on why 1 mM and 2 mM Mn(II) resulted in lower level of fluorescence (Figure 3B) – I believe this is because high concentrations of Mn(II) cause fluorescence quenching. This can be confirmed with the PAGE data proposed in point 3 above.

5) I am not sure if the authors are aware of the fact that Mn(II) precipitates at pH above 8 (the solution turns dark). This should be noted in Figure S5 as the data showed that lower fluorescence was observed with pH 8.5 and pH 9.0 than with pH 8.0.

6) Kinetic experiments shown in Figure S3 are not acceptable. More time points need to be taken between 3 seconds and 30 seconds to make the kinetic analysis more meaningful.

7) DNAzymes rarely cleave its substrates to entirety (mostly to 80-90% at most), as reported in nearly all prior RNA-cleaving DNAzyme studies (DNAzymes, nucleic acid enzymes in general, are known to misfold); however, the data in Figure S3 showed that the substrate cleavage reached 100%, suspicious of RNase-mediated cleavage which is known to cleave RNA substrates to entirety. Therefore, I suggest that the authors take necessary measures to rule out this possibility (autoclaving their reagents; performing some experiments with protease treated reagents; adding surrogate RNA molecules, such as commercially available tRNAs, in the cleavage reaction - the cleavage reaction mediated by the DNAzyme would not be slowed down by the surrogate RNA but the RNase mediated cleavage would).

Our revised manuscript has included new results (13 new Supplementary figures and 3 Supplementary videos), additional controls, and supporting experiments. These results rule out the possibility that RNases might mediated the cleavage. Our new experiments and Supplementary videos also provide a better characterization of our in vivo cell imaging and of the walking traces of our nanomachine. The revised manuscript addressed all comments and questions raised by the three reviewers.

Point-by-point responses are summarized below, with **revisions highlighted**.

Reviewer #1 (Remarks to the Author):

This paper reports the operation of an artificial nanomotor system inside living cells. The motor system is activated in the cells by a miRNA target together with a cofactor (Mn²⁺). The subsequent motor operation releases many copies of quenched 'signal' strands so that the amplified fluorescence serves as an effective indicator of the presence of the target miRNA. Artificial nanomotors are believed to have many potential applications inside living cells, but progress in this direction has been hard to come by. This study, though still a demonstration of principle, represents an important step forward and will likely inspire more to follow in future. I think that this paper is worth publication in Nature Communications after some issues are addressed or clarified (see below).

(1) The paper show enough experiments to prove the motor's in vitro operation (e.g., effectiveness of oligonucleotide target/cofactor activating the motor, fluorescence amplification, etc.). But evidence for the motor's in vivo operation is not as strong. For example, the FAM light spots form rings around a cell. This pattern is seen throughout Figs. 5-8 and especially clear in Fig. 6. It raises a question whether the light spots are produced by the Au-motor systems that remain outside the cells and are stuck on their surface. By simple logic, the light spots produced from an intracellular cause should spread all over a cell instead of forming a ring along the cell's edge. One more control experiment is necessary to rule out this possibility.

(2) Following (1), the paper briefly mentions an ICP-MS experiment to measure Au concentration inside the cells (Fig. S9). Is the experiment done using the Au particle with the DNA motors or without? How does the method distinguish between the Au-motor systems inside the cells from those stuck on the surface (outside)? Judging simply from the optical images, even some light spots are seemingly inside a cell, but they can be on top of the cell (outside it).

(3) A counterargument to (1) is perhaps that the miRNA trigger is present only inside the living cell. Any possibility of the miRNA leaking from the cells? What's the miRNA concentration in the cell, roughly?

Response to these three questions:

We thank the reviewer for the constructive remarks. We have included new experimental results to address these three related questions. Additional results include (i) a video showing fluorescence increases in the cells over 60 min, (ii) parallel negative controls built with a mutant DNAzyme, (iii) control experiments ruling out the issue of DNAzyme motor adsorption on the cell surface, and (iv) a control experiment ruling out the effect of miRNA leaking from the cells.

Response to question (1):

We have conducted a “control experiment to rule out the possibility” of adsorption. We have shown the results in the new Supplementary Figure S25, with description of the control experiment illustrated in Figure S24. On pages 21-22, we have added the follow clarification:

“To test whether possible adsorption of the DNAzyme motor on the cell surface could produce fluorescence signals that would confound the detection of intracellular targets, we conducted the following control experiment (as schematically depicted in Fig. S24). We first incubated MDA-MB-231 cancer cells with an inactive DNAzyme motor system constructed with the mutated DNAzyme. We then added 0.5 mM Mn²⁺, incubated for 60 min, and obtained fluorescence images of the cells. No fluorescence signal was detected (Fig. S25b), which is as expected because the mutated DNAzyme is not able to cleave the substrate. After washing the cells 3 times with 1×PBS, we further incubated the cells with 200 pM free control DNAzyme sequence and then measured cellular fluorescence after

incubation for additional 20 min. If the DNAzyme motor system were adsorbed on the surface of the cells, the free control DNAzyme would hybridize to the substrate strands and cleave them off from the DNAzyme motor system, generating fluorescence signal. Our results showed no detectable fluorescence (Fig. S25c), ruling out possible interference from potential adsorption of the DNAzyme motor on the cell surface...”

We have also provided Supplementary Video 2 and representative frames of cell images detected over 60 min of the motor operation (new Supplementary Fig. S18). We have included the whole set of images in Figure 6. We have also added images from monitoring fluorescence of the cells for a longer time, up to 5 hours (Figure S20). These multiple cell images complement our original data, supporting the intracellular operation of the motor.

Response to question (2):

We have now clarified that the experiments were done using the AuNP with the motors. We have added a schematic, Figure S23, to help describe the experiments, and we have revised the text accordingly on page 20:

“The cellular uptake of the DNAzyme motor system, consisting of AuNPs functionalized with substrate and locked DNAzyme sequences, was determined by using inductively coupled plasma mass spectrometry (ICP-MS) (Fig. S23)...”

Response to question (3):

We agree with the reviewer’s suggestion and have added the following control experiment to address the issue of possible leaking of miRNA from the cells. Our results support our original conclusion. We have added the following description on page 22:

“To examine whether the target miRNA could leak out of the cells and then initiate operation of the DNAzyme motor outside of cells, we conducted the following control experiment. We added a DNAzyme reaction buffer, containing 25 mM Tris-acetate (pH 8.0) and 125 mM NaCl, to the MDA-MB-231 cells, and removed the buffer either 1 h after incubation with the cells or immediately after its contact with the cells (1 min). We then added 230 pM of the DNAzyme motor and 0.5 mM MnCl₂ to these reaction buffer solutions and monitored fluorescence for 60 min (Fig. S26). If the target miRNA were leaked out of the cells, they would initiate operation of the DNAzyme motor and generate fluorescence signals. However, there was no detectable fluorescence increase (Fig. S26), suggesting that leaking of the target miRNA from the cells into the reaction buffer was negligible. As a positive control, further addition of 200 pM target miRNA to the solution resulted in an expected fluorescence increase (Fig. S26)”.

(4) Accepting that the Au-motor systems are inside, a question still remains whether the effect is triggered by a single species as the miRNA or other factors. Can the intracellular presence of a target DNA be ruled out? Most of in vitro tests are done using the target DNA instead of the miRNA (miR-10b). I wonder, why don't use the miRNA throughout the in vitro and in vivo experiments?

Response to question (4):

Our new control experiments using a mutant DNAzyme motor system, our results from both the target miRNA and the target DNA sequence, additional results on the concentration dependence of fluorescence intensity, and the results from the negative control cells can address these comments.

We constructed a mutant DNAzyme motor system which differed from the functional DNAzyme motor system only by changing two G’s to T’s in the catalytic core of the DNAzyme sequence. The mutated sequence lost its catalytic ability to cleave the substrate. Under the same operating conditions, the mutant DNAzyme motor system did not generate detectable fluorescence from the target cells (new Figure 5c, new Figure S20b, and new Figure S25b) or from solution (Figure 2b, Figure 4).

In addition to our original data where “Most of in vitro tests are done using the target DNA”, we have now added the results from the use of “the miRNA throughout the in vitro and in vivo experiments”.

“Having optimized the operating conditions of the DNAzyme motor responding to the DNA target, we further examined the operation of the DNAzyme motor in response to various concentrations of the target miRNA.” We have included a brief discussion on the motor response to both the target miRNA (Fig. 4b and Fig. S11) and DNA (Fig. S12). (on page 9).

(5) What's the zero of the operating time? Is it the time to add Mn^{2+} ? There seems to be no control over the timing of miRNA inside the cells.

Response to question (5):

The reviewer is correct; the zero of the operating time is the time when Mn^{2+} is added. The cofactor Mn^{2+} is required to activate the motor. We have now clarified in the manuscript and in figure captions that time 0 refers to when the cofactor Mn^{2+} is added.

We have also included a control mutant DNAzyme motor system in our parallel experiments. Please see response to Reviewer 2.

(6) How are the motor's steps determined (Fig. S7)?

Response to question (6):

We have now clarified how the motor's steps are determined, and included the description in the captions of new Fig. S13 (previously Fig. S7).

“The moving steps were estimated from measuring the fluorescent substrate segment **F1**. The concentration of the fluorescent substrate segment **F1** was determined against a calibration curve that was constructed by using standard solutions of FAM-labeled substrate. For example, from the calibration, the concentration of the fluorescent substrate segment **F1** after 27 min of the motor operation was determined to be 6.2 nM. This was initiated by 200 pM target miRNA. Because the concentration of miRNA is lower than that of the DNAzyme motor, each target miRNA molecule activates a single DNAzyme motor. Each walking step of the motor generates a substrate segment **F1**. Therefore, the detected overall 6.2 nM substrate segment **F1** is a result of 31 average walking steps of each DNAzyme motor initiated by 200 pM (0.2 nM) target miRNA ($6.2 \text{ nM}/0.2 \text{ nM} = 31$).”

(7) 'Amplified imaging of the intracellular microRNA target' appears not accurate. It's not a real imaging since the amplified FAM emission signals only the presence of the miRNA, but not its whereabouts. The miRNA location is still from Cy5 emission, which is independent of the motor-induced FAM amplification.

Response to question (7):

We have changed “amplified imaging” to “amplified detection” (e.g., on page 1, line 12).

Reviewer #2 (Remarks to the Author):

In this manuscript, the authors described strategy of constructing a DNAzyme motor system on a gold nanoparticle (AuNP), which has four features. Integration of all motor components onto a single AuNP facilitates cellular uptake of the DNA motor system. They found that this system could allow real-time imaging of the intracellular operation of the motor.

The work is interesting but the following concerns need to be addressed.

1. The description of the concept of DNAzyme motor system is too brief. I am not sure if any random non-oriented walking without tracking could be defined as "motor". Also it is better if the authors could provide a tracking video for the fluorescent molecules releasing from AuNPs.
4. Except monitoring of the fluorescent molecules detached from the AuNP in real-time imaging, could you also track

its walking traces? Because even without the release of fluorescence dye, the AuNPs were continually randomly diffusing from here to there. Also, was its walking orientated and rational designed?

7. Can you track each fluorescent molecule cleaved by DNAzyme which might indicate the exact position of your AuNP, even so, it is also difficult to distinguish which dye molecule belongs to which AuNP. The increase value of fluorescence intensity is a collective behavior.

Responses to questions 1, 4, and 7:

We thank the reviewer for the constructive suggestions. We have now added two sets of videos, tracking the operation of the DNAzyme motors on individual AuNPs (Supplementary Video 1) and detecting the intracellular operation of the DNAzyme motors (Supplementary Video 2).

We agree with the reviewer that “the AuNPs were continually randomly diffusing from here to there.” We have also clarified in the manuscript: “Like other DNA motors that use DNA tracks built on nano- and micro- materials^{8, 30, 31}, walking of the DNAzyme motor along the AuNP is stochastic.” (page 5).

The reviewer correctly recognized that “it is difficult to distinguish which dye molecule belongs to which AuNP”, and therefore it is difficult to track the walking traces. To overcome this problem, we have modified the system to enable us to obtain tracking videos. We have include a schematic to show the design (Figure S14), representative images from the videos (Figure S15), Supplementary Video 1a (in the presence of the target miRNA) and 1b (in the absence of the target miRNA), and the following descriptions on page 10:

“To trace the operation of the DNAzyme motor on individual AuNPs, we designed an alternative system that enables each walking step of the motor to turn on the fluorescence of a Cy5 molecule on the AuNP (Supplementary Fig. S14 and Supplementary Table 3). We designed the substrate strand to have a hairpin structure with a long single-stranded overhang that hybridizes to a Cy5-labeled DNA strand. The end of the substrate strand without the overhang is labeled with a black hole quencher so that hybridization of the substrate strand to the Cy5-labeled strand quenches the fluorescence of Cy5. We conjugated onto each AuNP dozens of locked DNAzyme strands and hundreds of Cy5-labeled strands to which the hairpin substrates are hybridized. In the presence of the target miRNA, the locked DNAzyme is activated to cleave the substrate at the single-ribonucleotide junction in the hairpin loop. The cleavage disrupts the hairpin structure and releases a quencher-containing fragment from the AuNP, restoring the fluorescence of the Cy5 molecule. The DNAzyme dissociates from the cleaved substrate and hybridizes to the next substrate, achieving the walking of the DNAzyme motor from one substrate strand to the next. Each walking step restores the fluorescence of a Cy5 molecule that remains attached to the AuNP. Our results (Supplementary Video 1a and Fig. S15a) indeed show that the fluorescence of Cy5 from individual AuNPs increases over time, representing the stepwise walking of the DNAzyme motor on the AuNP and the corresponding catalytic cleavage of the quencher-labeled substrate. In the absence of the target miRNA, there is very little fluorescence background (Supplementary Video 1b and Fig. S15 b), consistent with the fact that the DNAzyme motor is inactive.”

2. In figure 5a, Images of live cells after uptake of a DNAzyme motor system were provided. I am wondering what will be like when you extend the reaction time from 1h to 3h, 5h. Is it possible that after longer time incubation, the fluorescent molecules will be released from the AuNPs?

Response to question 2:

We have added Figures S20a and S20b to address this question. When we extend the reaction time from 1 h to 3 h and 5 h, we observed fluorescence in the cells as “a result of intracellular operation of the DNAzyme motor initiated by the target microRNA (miR-10b)... The slight decrease of fluorescence intensity observed after 5 h is probably due to photo bleaching of the fluorescent substrate.” (Figures S20a). We also conducted parallel experiments using the mutant DNAzyme motor. Figures S20b shows “Images of MDA-MB-231 cancer cells after incubation with the mutant

DNAzyme motor for 2 h followed by the addition of Mn^{2+} . Fluorescence imaging at 0, 1, 3, and 5 h after the addition of Mn^{2+} shows only background, indicating that the substrate strands on AuNPs are stable in the cells.” These results show that fluorescent molecules are not released from the AuNPs without the action of the functional DNAzyme motor.

3. In figure S12, why was the fluorescence decreased for the miRNA initiated DNAzyme motor after 3 h. The authors claimed it is because of the photo bleaching, but did the control DNAzyme motor system undergo the same condition of photo bleaching? Could you please explain it more?

Response to question 3:

We have added another control experiment to Figure S21 (Figure S12 in the original manuscript) to further clarify the small fluorescence decrease for the miRNA-initiated DNAzyme motor after 3 h. We included the following description in the caption of Figure S21:

“The small decrease of fluorescence from the target-initiated DNAzyme motor after 3 h is probably due to photo bleaching by extended period of excitation light. Support for this is from the following experiment. 3 h after the operation of the free control DNAzyme, EDTA was added to the solution to chelate the cofactor Mn^{2+} and thus stop the further operation of the DNAzyme. Continued monitoring of the fluorescent substrate fragment produced during the first 3 h by the free DNAzyme also shows the same slight decreasing pattern as in the case of the target-initiated DNAzyme motor.” Therefore, in both the control experiment and in the target-initiated DNAzyme motor experiment, when there is no further generation of new fluorescent substrate molecules after 3 h, the fluorescence decreases similarly because both undergo the same condition of photo bleaching.

5. Could you provide the probe density of both substrate strand and DNAzyme strand on AuNP?

Response to question 5:

We have now provided the probe density of both substrate strand and DNAzyme strand on AuNP. We added the following description on page 17:

“On average, 232 substrate molecules were conjugated to each AuNP. On the basis of a molar ratio of 1:20 for the DNAzyme strand and the substrate strand used together in the conjugation reaction, approximately 12 DNAzyme molecules were conjugated to each AuNP. Thus, the densities of the substrate and DNAzyme strands are $1.85 \times 10^{-1}/nm^2$ and $9.24 \times 10^{-3}/nm^2$ on the 20-nm AuNPs”.

6. How did authors define the moving steps in Figure S4 and Figure S7? What is the relationship between each moving step and fluorescence increasing?

Response to question 6:

We have estimated the moving steps and have shown the results in Figures S9 and Figure S13 (Figures S4 and S7 in the original manuscript).

We have now described how we estimated the moving steps in the captions of new Fig. S13:

“The moving steps were estimated from measuring the fluorescent substrate segment **F1**. The concentration of the fluorescent substrate segment **F1** was determined against a calibration curve that was constructed by using standard solutions of FAM-labeled substrate. For example, from the calibration, the concentration of the fluorescent substrate segment **F1** after 27 min of the motor operation was determined to be 6.2 nM. This was initiated by 200 pM target miRNA. Because the concentration of miRNA is lower than that of the DNAzyme motor, each target miRNA molecule activates a single DNAzyme motor. Each walking step of the motor generates a substrate fragment (**F1**). Therefore, the detected overall 6.2 nM substrate segment **F1** is a result of 31 average walking steps of each DNAzyme motor initiated by 200 pM (0.2 nM) target miRNA ($6.2 \text{ nM}/0.2 \text{ nM} = 31$).”

8. Why each AuNP contained only a single activated DNAzyme motor when the target sequence concentration was less than 230 pM? How did you quantify the DNAzyme motors in the cells?

Response to question 8:

We have clarified this question by revising the sentence in the original manuscript, now on page 9:

“We have determined that on average approximately 12 locked DNAzyme motors were conjugated on each AuNP. With a concentration of 230 pM AuNPs used in the operation, the total number of the DNAzyme motors are in large excess over the target miRNA when the concentrations of miRNA are lower than that of AuNPs. Therefore, only one (or none) of the DNAzyme motors on each AuNP is activated by the target miRNA.”

We have also clarified how we quantified the DNAzyme motors in the cells and added a new Figure S23 to depict the experimental procedures. Please see descriptions on page 20:

“The cellular uptake of the DNAzyme motor system, consisting of AuNPs functionalized with substrate and locked DNAzyme sequences, was determined by using inductively coupled plasma mass spectrometry (ICP-MS) (Fig. S23)...”

Reviewer #3 (Remarks to the Author):

This manuscript reports the design of gold nanoparticle/RNA cleaving DNAzyme based nanomachine for the detection of microRNAs in cells. Such a system has not been reported before and the approach is definitely novel and could open up potential applications for cellular imaging of microRNA and other targets. Overall the study was reasonably well designed and executed, and the manuscript is well written and easy to follow. However, I do have several concerns, which need to be addressed before I can recommend its publication in Nature Communications.

1) My biggest concern is: Is the DNAzyme actually driving the operation of the nanomachine? Most of the data presented seem to argue for this key claim of the work, but I am not totally convinced because there is a great possibility that RNases might have actually mediated the RNA cleavage reaction. RNases are ubiquitous and potent enzymes. Many research agents (buffers, salts, water) are contaminated with RNases and cells contain all kinds of RNases. Therefore, I am concerned about the validity of the claim. An important experiment that the authors must perform is the use of a mutant DNAzyme where the catalytic core of the DNAzyme is mutated from TCCGAGCCGGTCGAA to TCCGA~~t~~CCGGT~~t~~CtAA. This mutant DNAzyme should be used as a control to perform the key experiments throughout the study (Figures 2B, 3A, 4, 5, 6 and 7). Without these experiments, the validity of the claims is questionable.

Response to question 1):

We thank the reviewer for the very helpful suggestion. As suggested, we have now included “a mutant DNAzyme where the catalytic core of the DNAzyme is mutated from TCCGAGCCGGTCGAA to TCCGA~~t~~CCGGT~~t~~CtAA”. (Sequence is shown in Table S1). We have used this “mutant DNAzyme as a control to perform the key experiments throughout the study”. The results are included in the new Figures 2b, 3a, 4, 5, S3, S5, S19, S20b, S24, and S25. The parallel results from the use of the functional DNAzyme and the mutant DNAzyme support validity of our findings.

We have taken necessary measures to eliminate RNases-mediated cleavage, e.g., by autoclaving the reagents and using RNase inhibitors.

Our additional control experiments also rule out the RNases-mediated cleavage. For example, in Figures S3, S5, S6, and S7, the far left lane of each gel image is from the full-length substrate in the autoclaved reaction buffer, containing 25 mM Tris-acetate and 200 mM NaCl, pH 8.0. No cleavage product is visible from this control, which contains substrate in the autoclaved reaction buffer. The cleavage product is only observed from the DNAzyme-mediated cleavage of the substrate.

2) Only the data in Figure 3B has error bars. Were replicates done for all the experiments? If so, the information should be provided and statistic analysis performed.

Response to question 2):

Yes, replicates were done for all the experiments. We have included error bars in all bar graphs. We have added relative standard deviations (RSDs) to the captions of other figures where error bars are either smaller than the size of the symbols or would be too crowded in the figure.

3) PAGE analysis for all the in vitro cleavage reactions (Figure 2, b, c, d; 3b, 4a and 4b) should be performed and the data should be provided in SI.

Response to question 3):

As suggested, we have performed PAGE analysis and included the data in Supplementary Information, new Figures S3, S4, S5, S6, and S7. The results of PAGE analysis are consistent with those of corresponding experiments using fluorescence detection.

4) The authors need to comment on why 1 mM and 2 mM Mn(II) resulted in lower level of fluorescence (Figure 3B) – I believe this is because high concentrations of Mn(II) cause fluorescence quenching. This can be confirmed with the PAGE data proposed in point 3 above.

Response to question 4):

We agree with the reviewer's suggestion. Indeed, we show that "high concentrations of Mn(II) cause fluorescence quenching". (Figure S8). We have also conducted PAGE analysis. As shown in new Figure S7, the band corresponding to the cleavage product **F1** appears later when 1 mM and 2 mM Mn^{2+} were used as compared to when 0.5 mM Mn^{2+} was used. We have added the following clarification:

"The slightly lower fluorescence intensity of the fluorescent substrate segment **F1** is probably because of a combination of the following: (i) the higher Mn^{2+} concentration decreased the cleavage rate (Fig. S7) by slowing down the dissociation of DNAzyme from the cleavage product **F2**; and (ii) a reduced fluorescence intensity of **F1** due to fluorescence quenching (Fig. S8)."

5) I am not sure if the authors are aware of the fact that Mn(II) precipitates at pH above 8 (the solution turns dark). This should be noted in Figure S5 as the data showed that lower fluorescence was observed with pH 8.5 and pH 9.0 than with pH 8.0.

Response to question 5):

Yes, we "are aware of the fact that Mn(II) precipitates" at high pH.

As suggested, we added a note to the caption of Figure S9 (Figure S5 in the original manuscript):

"Note that at higher pH (e.g., >8.5), Mn^{2+} could precipitate as $Mn(OH)_2$, according to K_{sp} of $Mn(OH)_2$ which is 1.9×10^{-13} ."

We did not observe a change of solution color, probably because the concentration of Mn^{2+} we used was low (0.5 mM). From the solubility product constant K_{sp} of $Mn(OH)_2$ (1.9×10^{-13}), at pH 9.1, the maximum soluble concentration of Mn^{2+} would be 1 mM, above which Mn^{2+} precipitates as $Mn(OH)_2$.

6) Kinetic experiments shown in Figure S3 are not acceptable. More time points need to be taken between 3 seconds and 30 seconds to make the kinetic analysis more meaningful.

Response to question 6):

As suggested, we have performed additional experiments and we now have four measurements between 3 seconds and 30 seconds, as shown in the new Figure S6 (Figure S3 in the original manuscript). The extra time points are good for the kinetic analysis.

7) DNAzymes rarely cleave its substrates to entirety (mostly to 80-90% at most), as reported in nearly all prior RNA-cleaving DNAzyme studies (DNAzymes, nucleic acid enzymes in general, are known to misfold); however, the data in Figure S3 showed that the substrate cleavage reached 100%, suspicious of RNase-mediated cleavage which is known to cleave RNA substrates to entirety. Therefore, I suggest that the authors take necessary measures to rule out this possibility (autoclaving their reagents; performing some experiments with protease treated reagents; adding surrogate RNA molecules, such as commercially available tRNAs, in the cleavage reaction - the cleavage reaction mediated by the DNAzyme would not be slowed down by the surrogate RNA but the RNase mediated cleavage would).

Response to question 7):

We thank the reviewer for the suggestions. Indeed, we have taken “necessary measures to rule out the possibility of RNase-mediated cleavage”. We autoclaved reagents and added commercially available RNase inhibitor (from Promega) to the cleavage reaction solutions. We also tested the addition of tRNA. Taking necessary measures as suggested, we have also repeated the experiments on the single-turnover cleavage rate of the substrate by the DNAzyme motor. Also in response to question 6), we have added four measurements between 3 seconds and 30 seconds. Our results are shown in the new Figure S6 (Figure S3 in the original manuscript).

We have also re-calculated the percentage of the substrate cleaved as described below. From these measures, we show cleavage of 80-90% of the substrate, which is consistent with the reviewer’s suggestion and consistent with literature reports.

“The sum of the intensity of the substrate band and the product band in the first lane at 0.05 min was used to represent the total amount of the fluorescent substrate and product, serving as the denominator in the calculation of the percentage cleaved. The intensity of the cleaved product band in each lane from 0.05 to 10 min was used as the numerator in the calculation of the percentage cleaved.”

We also included negative controls – the substrate in the reaction buffer. For example, in Figures S3, S5, S6, and S7, the far left lane of each gel image is from the full-length substrate in the autoclaved reaction buffer (pH 8.0), containing 25 mM Tris-acetate and 200 mM NaCl. No cleavage product is visible from this control, suggesting that the substrate is stable in the autoclaved buffer. The cleavage product is only observed from the DNAzyme-mediated cleavage (e.g., bottom gel images from Figure S3 and S5).

Reviewer #1 (Remarks to the Author)

My problems are well answered. I recommend publication of the paper in its present form.

Reviewer #2 (Remarks to the Author)

The authors have addressed my comments. I recommend it for publication.

Reviewer #3 (Remarks to the Author)

I am pleased with the revised manuscript as the authors have made very significant effort to revise their manuscript to address my concerns. It is now a really nice manuscript and I support its publication in Nature Communications.